# microRNA-184 distribution and consequences on glial septate junctions and the blood-brain barrier

**Sravya Paluri, Vanessa J. Auld** [ID]*

Department of Zoology, Life Sciences Institute, University of British Columbia, Vancouver, British Columbia, Canada

* auld@zoology.ubc.ca

## Abstract

Cellular permeability barriers restrict the diffusion of solutes, pathogens, and cells across tissues. In *Drosophila melanogaster*, septate and tricellular junctions create permeability barriers in epithelia and the blood-brain barrier in glia. *In vitro*, and *in vivo* studies in the epithelia of the *Drosophila* wing imaginal discs identified microRNA-184 (miR-184) as a potential regulator of a subset of pleated septate and tricellular junction proteins. However, which tissues express miR-184, and the consequences of miR-184 expression on the blood-brain barrier has not been examined. Using a miR-184 sensor, we found that miR-184 is absent in tissues with pleated septate junctions but is present in tissues with smooth septate junctions. When expressed in the subperineurial glia that form the blood-brain barrier, miR-184 resulted in the loss of targeted septate junction proteins, a compromised blood-brain barrier, decreased locomotion, and lethality. Interestingly, qRT-PCR analysis revealed that miR-184 expression did not alter mRNA levels of targeted genes. Conversely, expression of miR-184 led to an increase in the mRNA and expression of the non-target Nervana2 protein. Thus, mRNA-184 can regulate multiple pleated septate junction proteins either directly through loss of translation or indirectly by disruption of the septate junction domain.

## Introduction

Permeability barriers in tissues like the epidermis, intestine, and brain are essential to prevent fluid flow and restrict the diffusion of molecules and pathogens across these tissues. This functionally conserved permeability barrier consists of protein complexes that underlie tight junctions (TJ) and septate junctions (SJ) in vertebrates and invertebrates respectively [1–4]. Septate junctions (SJs) have two distinct types: smooth (sSJ) and pleated (pSJ), which are constituted by different protein complexes. sSJs are formed by mostly endodermally-derived epithelia including the

**Data availability statement:** All relevant data are within the paper and its Supporting information files.

**Funding:** V.A AWD-010749 Natural Sciences and Engineering Research Council.

**Competing interests:** The authors have declared that no competing interests exist.

midgut and gastric caeca, and some ectodermally-derived tissues like the Malpighian tubules, while pSJs are found in ectodermally-derived epithelia that include the epidermis, nervous system, salivary glands, foregut, hindgut, trachea, and imaginal discs [5–7].

The *Drosophila* pSJ consists of a core of interdependent proteins including the highly conserved proteins NeurexinIV (Nrx-IV), Neuroglian, Contactin, Coracle, the Na/K-ATPase alpha and beta-subunit Nervana2 (Nrv2), α-2-macroglobulin family protein Macroglobulin complement-related (Mcr) and the Claudin-like proteins Megatrachea, Kune-kune (Kune) and Sinuous (Sinu) [3,8–15]. A second permeability barrier is created at the tricellular junction (TCJ), which is formed by the convergence of pSJs from three neighboring cells [16–18]. The TCJ is created by a protein complex that includes Gliotactin (Gli), Bark beetle (Bark), also known as Anakonda, and M6 [18–20].

Disruption of pleated septate junctions causes embryonic lethality due to defects in dorsal closure, epidermal cuticle, and trachea size [10,14,21–23]. Loss of pSJ and TCJ proteins also leads to loss of embryonic lethality due to loss of permeability barriers in the salivary gland, tracheal, blood-brain and blood-nerve barriers [3,9,22,24]. Of interest, the blood-nerve and blood-brain barriers are created by the subperineurial glia (SPG) through the formation of pSJ along the edge of a single SPG cell and between adjacent SPG [3]. While the components of the pSJ are well-defined in many tissues, how the complex of pSJ proteins is regulated is poorly understood.

One potential mechanism to regulate the pSJ and TCJ complex is through microRNAs (miRNAs), which can regulate the expression of cohorts of proteins through control of mRNA stability or translation. A strong candidate to regulate the pSJ protein complex is microRNA-184 (miR-184). miR-184 is predicted to target 85 mRNAs (targetscanfly.org) including a suite of 10 pSJ mRNAs (for example: Nrx-IV, Mcr, Sinu, Kune, and TCJ mRNAs, Gli, M6), many of which were verified *in vitro* [25]. Of these, Nrx-IV, Mcr and Gli were confirmed as targets *in vivo* in the wing imaginal disc [26]. Thus, miR-184 may be targeting key pSJ and TCJ proteins in multiple tissues.

To test for the distribution of miR-184, we designed a miR-184 sensor expressing a yellow fluorescent protein (YFP) containing miR-184 binding sites in the 3'UTR region, rendering it sensitive to miR-184. We observed the presence of miR-184 in multiple tissues including imaginal discs, muscles, and regions of the intestinal system known to express smooth SJs. Conversely miR-184 was absent from multiple tissues known to express pleated septate junctions such as body wall epithelia, the hindgut and the subperineurial glia that create the blood-brain barrier. These same tissues express the predicted miR-184 pSJ and TCJ targets, and expression of miR-184 in the subperineurial glia disrupted the pSJ and the blood-brain barrier but without change mRNA levels of the predicted target proteins. Overall, we found that miR-184 levels in pSJ tissues are likely tightly regulated given that expression leads to loss of pSJ proteins, loss of the blood-brain barrier, larval lethality, and disruption of animal locomotion.

## Materials and methods

### Drosophila strains

The following lines were used: Gli-GAL4 [27], SPG/moody-GAL4 [28], Mdr65-Gal4 (Bloomington Stock Center), Gli-lacZ (P{PZ}Gli^rL82, Bloomington Drosophila Stock Center), Tubulin-LexA (Bloomington Drosophila Stock Center), UAS-mCD8::GFP [29], UAS-mCD8::RFP (Bloomington Drosophila Stock Center), UAS-miR184 (FlyORF), UAS-lacZ [30], $w^{1118}$, Bark::GFP [31], Nrx-IV::GFP [32] and UAS-Nrx-IV-RNAi (GD2436; Vienna Drosophila Stock Center). All crosses were reared at 25°C unless specified and control crosses were Gli-Gal4 or SPG-Gal4 crossed with w[1118]. For the expression of UAS-miR-184, the experimental lines were embryonically lethal when raised continuously at 25°C, thus expression was titrated by raising at 18°C until the first- instar larval stages and shifting to 25°C until the third-instar larval stage.

### Immunolabelling and microscopy

Third-instar wandering larvae were dissected, fixed, and stained for antibodies as previously described [33]. Primary antibodies used were: rabbit anti-Kune (1:300) [13], mouse anti-Gli (1:300) [24], rat anti-Mcr (1:100) [12], rabbit anti-Sinu (1:50) [15], guinea pig anti-M6 (1:250) [31], rabbit anti-Nrv2.1 (1:1000) (Invitrogen), mouse anti-Repo (1:10) (Developmental Studies Hybridoma Bank) and mouse anti-Discs large (1:100) (Developmental Studies Hybridoma Bank). Secondary antibodies (1:300) were goat anti-rabbit, goat anti-mouse, goat anti-rat, and goat anti-guinea pig conjugated with Alexa 568 or Alexa 647 (Invitrogen). After mounting in Vectashield (Vector Laboratories), fluorescent images of the slides were taken either using a 60x oil immersion objective (NA 1.4) on the Deltavision microscope (GE Healthcare) with 0.2 μm slices or using a 20X oil immersion objective (NA = 0.7) on the Leica-SP5-inverted microscope (Life Science imaging (LSI) facility, UBC) with 1 um steps. The 60X images generated from the Deltavision microscope were then deconvolved using the software SoftWorx (SoftWorx) using a point-spread function measured with a 0.2 μm fluorescently labeled bead in Vectashield (Vector Laboratories). Images were imported into Fiji software [34]. For images of peripheral nerves, a single Z stack was used, and for brain lobes, a 'Z projection' across the pSJ was used. The final figures were compiled using Adobe Photoshop (Adobe Systems).

### LexAOP-NLS-YFP-184 sensor

The miRNA-184 binding sequence 5'CCGTCCA3' was utilized from the 3'UTR of the Nrx-IV mRNA. Two copies of the miRNA-184 binding sites separated by a random 9 bp of linker sequence 5'-ttttaaata-3' and flanked by homologous sequences, were inserted into 3'UTR region of p{attbYFP} pHStinger (NLS-YFP) (RRID: DGRC_1018) by seamless cloning by using the primer:

5'-gaacgaagacgctaaactagaata**ccgtcca**ttttaaata**ccgtcca**ttttatatggctgattatga-3' (with miR-184 target sequences in bold). The NLS-YFP plus miR-184 binding sites was placed under the control LexAOP. To acquire the backbone of LexAOP plasmid, the myr::GFP fragment was released from the pJFRC19–13XLexAop2-IVS-myr::GFP [35] (RRID: Addgene_26224) and replaced with NLS-YFP containing miR-184 binding sites. The LexAOP::NLS-YFP plasmid generated was integrated into the second chromosome and third chromosome in two separate injections at the attp40 site and the attp2 site respectively using the PhiC31 integrase system (Bestgene Inc.) to generate LexAOP-NLS-YFP fly lines on the second and third chromosome respectively.

### Imaging quantification

For quantifications, a single brain lobe was analyzed per larvae whereas for the peripheral nerves, the A1-A8 nerves were analyzed, N = number of larvae unless specified otherwise. For Kune fluorescence intensity measurements, while imaging the exposure conditions for each laser were maintained the same across brain lobes and peripheral nerves. Images were processed in Fiji [34] by generating a Z stack of the 'sum' of the slices. The mean fluorescence intensity was measured

across a region of interest of 142 square pixels for Kune and normalized against the intensity of Nrx-IV in the same area. For the CNS, three different areas along the pSJ area of the brain lobe were measured and averaged to determine the mean intensity per brain lobe. Each brain lobe analyzed was from a different larva. In the peripheral nerves, the fluorescent intensity was measured for 3–4 nerves of the A4-A8 region, which was then averaged to determine the mean intensity of peripheral nerve per animal. For Nrv2 fluorescence intensity measurements larvae were immunolabelled with the Nrv2.1 antibody. 3–4 A4-A8 peripheral nerves per larvae were imaged under the same exposures and deconvolved under the same conditions for all genotypes. Single z-slices with pSJ (or remnants) were analyzed using Fiji software. Three regions of interest of 84-pixel square area each spanning the width of each nerve were measured and averaged to determine the average fluorescence intensity of Nrv2 per animal.

### Larval locomotion assay

Locomotion assays were performed as previously described [36]. We used a Canon VIXIA HF R800 video camera for recording the larval locomotion videos for 1 minute and files were converted from *.MOV to *.AVI files by FFmpeg (ffmpeg. org). Three different parameters of locomotion (average speed, maximum speed, and total length traveled) were assessed using the wrMTrck plug-in in Fiji [37,38].

### Dye penetration assay

The assay was performed on brain lobes of third-instar larvae as previously described [39]. Larvae were bathed in 2.5 mM of a fixable 10,000 MW of Dextran conjugated to Texas Red dye in Schneider's medium (Invitrogen, D1836, Lysine Fixable). After standard fixation and washes [33], larvae were mounted in Vectashield. Larvae were imaged using a Leica-SP5-inverted microscope (Life Science imaging facility, University of British Columbia, Vancouver) using a 20X oil immersion objective (NA = 0.7) with 1 μm steps, using a laser at 561nm, 31% intensity, 612V smart gain and 0% offset for all the genotypes. After imaging, a single Z-slice of the images was used for the analysis of the fluorescent intensity. The 'Fire Look Up' table generated pseudo-colored images to reflect the intensity differences clearly between different areas of the image (ImageJ). Total fluorescence intensity was measured in the neuropile region for each brain lobe per animal.

### qRT-PCR analysis

The nervous system (including brain lobes, ventral nerve cord and peripheral nerves) of ~30 larvae of *SPG-Gal4 > UAS-lacZ* (control) and *SPG-Gal4 > UAS-miR-184* (experimental) were dissected for three different biological replicates. Total RNA was isolated using TRIzol reagent (ThermoFisher Scientific) and cDNA libraries were generated using 950 μg of total RNA with random primers using QuantiTect Reverse Transcription Kit (Qiagen). qRT-PCR was conducted using SYBR Green (ThermoFisher Scientific) with pSJ, TCJ primers (listed below), and actin primers as the control. To compare the mRNA levels, the ΔΔCt (threshold cycle) value was determined by the Livak comparative Ct method [40]. The quantification cycle values (Cq; the number of PCR cycles undergone to reach the threshold fluorescent value) for each primer/mRNA were measured and normalized with the corresponding actin values to get the ΔCt value. To get the ΔΔCt value for each primer/mRNA, the ΔCt control value (*UAS-LacZ*) was subtracted from the experimental value (*UAS-miR184*). The different primers used were:

| Actin | Forward primer | 5'-TTGTCTGGGCAAGAGGATCAG-3' |
|---|---|---|
| | Reverse primer | 5'-ACCACTCGCACTTGCACTTTC-3' |
| Bark | Forward primer | 5'-CGATAATGCGCCATCCACAAAGGC-3' |
| | Reverse primer | 5'-GGCATTGAATCCTGCACCACTTCC-3' |
| Gli | Forward primer | 5'-CTGGCCTCGTTCTACGACGTGG-3' |
| | Reverse primer | 5'-GAGGATTCCGTAGTTTCCGGGC-3' |

| Kune | Forward primer | 5'-GGCTGGTTACAGACGGCAGATTAC-3' |
|---|---|---|
| | Reverse primer | 5'-CGGTGTATGTCCTGGAAGTTGTTG-3' |
| M6 | Forward primer | 5'-CTCCTGGGAGTAGGAATCTTCTGC-3' |
| | Reverse primer | 5'-CATATCAGGCGTAGGTGGAAAACCTG-3' |
| Mcr | Forward primer | 5'-GCGTACTTCTTCACAACGGATCCG-3' |
| | Reverse primer | 5'-GCGGATCGTGGCCTTGATTGTG-3' |
| Nrv2 | Forward primer | 5'-CCCAACACGAGAACTACAAGCAC-3' |
| | Reverse primer | 5'-GTTGTAGATGTTCTGACCACGGC-3' |
| Nrx-IV | Forward primer | 5'-GTGATGAAAGACAGCACTGAACTGG-3' |
| | Reverse primer | 5'-GACGAACGACGAACGGGTAGTG-3' |
| Sinu | Forward primer | 5'-CCCACTTCGCACGCGAGGATTC-3' |
| | Reverse primer | 5'-CCAACGCCACAACTTCCGGATAG-3' |

## Statistical analyses

All statistics were performed using Prism 9.0 (GraphPad Software Inc). A Shapiro-Wilk test was performed ($\alpha = 0.05$) to test for normality of distributions. Both Mann-Whitey test (non-parametric T-test) or a Kruskal-Wallis test (non-parametric ANOVA) with Dunn's multiple comparisons test between groups were used to determine statistical significance. For more details on the statistical analysis, refer to the figure legend.

## Results

### microRNA-184 is differentially expressed across multiple tissues

To determine the range and distribution of miR-184 expression, we designed a miR-184 YFP sensor. For this, we inserted two miR-184 binding sites taken from the Nrx-IV gene (S1 Fig) and placed them in the 3'UTR region of a YFP transgene that contains a nuclear localization signal (NLS). The presence of miR-184 prevents translation of the NLS-YFP mRNA by either degradation or translational repression, resulting in no nuclear YFP localization. Alternatively, if miR-184 is absent, the NLS-YFP is translated and localized to the nucleus. The transgene was placed under the control of LexAOP (*LexAOP-NLS-YFP*), and we crossed the resulting lines to a ubiquitous lexA driver, *tubulin-lexA,* to profile miR-184 expression in different tissues.

We observed strong NLS-YFP in the nucleus of trachea (yellow arrowhead, Fig 1A), salivary glands (yellow arrowhead, Fig 1B), fat bodies (yellow arrowhead, Fig 1C), and body wall epithelia (yellow arrowhead, Fig 1D), indicating that these tissues do not express miR-184. We also found tissues with no NLS-YFP expression, such as body wall muscles (white arrow, Fig 1E), while other tissues had a mix of YFP-positive and -negative cells. In particular, while the majority of imaginal discs (n = 20 larvae) had no YFP expression (Fig 1F), some wing discs had small clusters of NLS-YFP-positive cells, which are presumably the adult muscle precursor cells [41]. This indicates that other than a few clusters of cells, most cells in the imaginal discs express miR-184.

Of interest was the differential expression patterns in the 3rd instar intestinal tract (Fig 2), which consists of tissues expressing both sSJs (midgut and Malpighian tubules) and pSJs (foregut and hindgut). We first analyzed the proventriculus, a pear-shaped region of the anterior midgut composed of the wall of esophagus surrounded by a layer of midgut epithelia [42]. We found that all the nuclei in the foregut and the wall of esophagus expressed NLS-YFP (yellow arrowhead, Fig 2A). However, in the surrounding wall of midgut around this region, there was no NLS-YFP expression (white arrow, Fig 2A). Thus, the proventriculus consists of a mixed population of cells with respect to miR-184 expression. We then analyzed the two tissues in the gut with sSJs: the midgut and Malpighian tubules. The midgut is compartmentalized into distinct regions [43], where we detected no NLS-YFP signal in the posterior midgut (white arrows, Fig 2B), but observed strong expression limited to a subset of cells in the middle midgut (yellow arrowhead, Fig 2B). Thus, we found that the

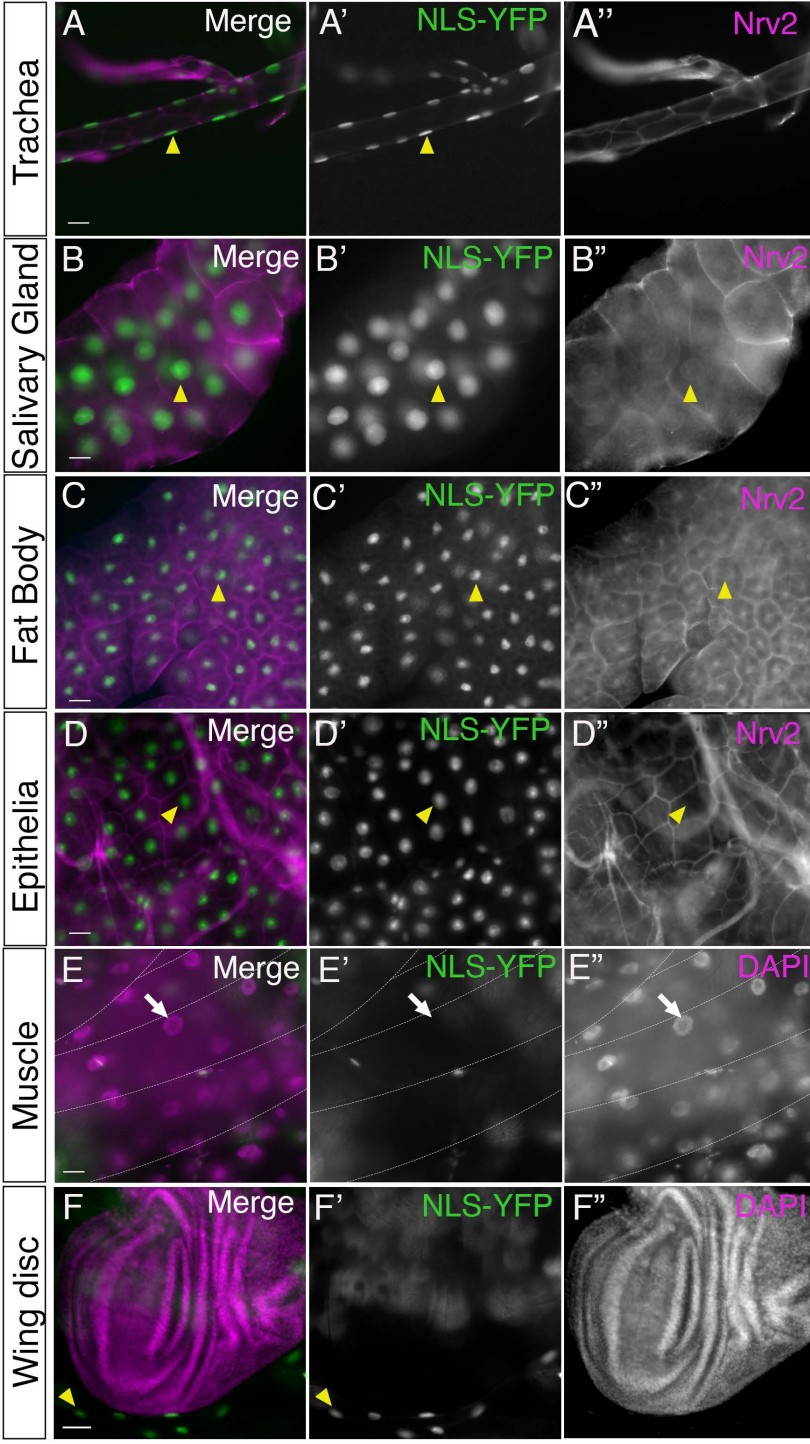

**Fig 1. miR-184 sensor expression across a range of tissues.** Different tissues isolated from third instar larvae of *tubulin-LexA* crossed to *LexAOP-NLS-YFP* (green) and immunolabelled with the nucleus marker DAPI (magenta) or the SJ marker Nrv2 (magenta). All images are projections of multiple Z sections. Nuclei were positive for NLS-YFP (yellow arrowheads) indicating the absence of miR-184 in the Trachea (A), Salivary gland (B), Fat Body (C) and body wall epithelia (D) (8 larvae). (E) Muscles: Lateral body wall muscles are indicated by the dashed white lines. No muscle nuclei assayed were positive for NLS-YFP, which were visualized with DAPI (white arrows), indicating the presence of miR-184 (6 larvae). (F) Wing imaginal disc: NLS-YFP was absent from the entire imaginal disc. NLS-YFP was expressed in neighboring trachea (yellow arrowhead) (6 larvae) Scale bars = 50 μm.

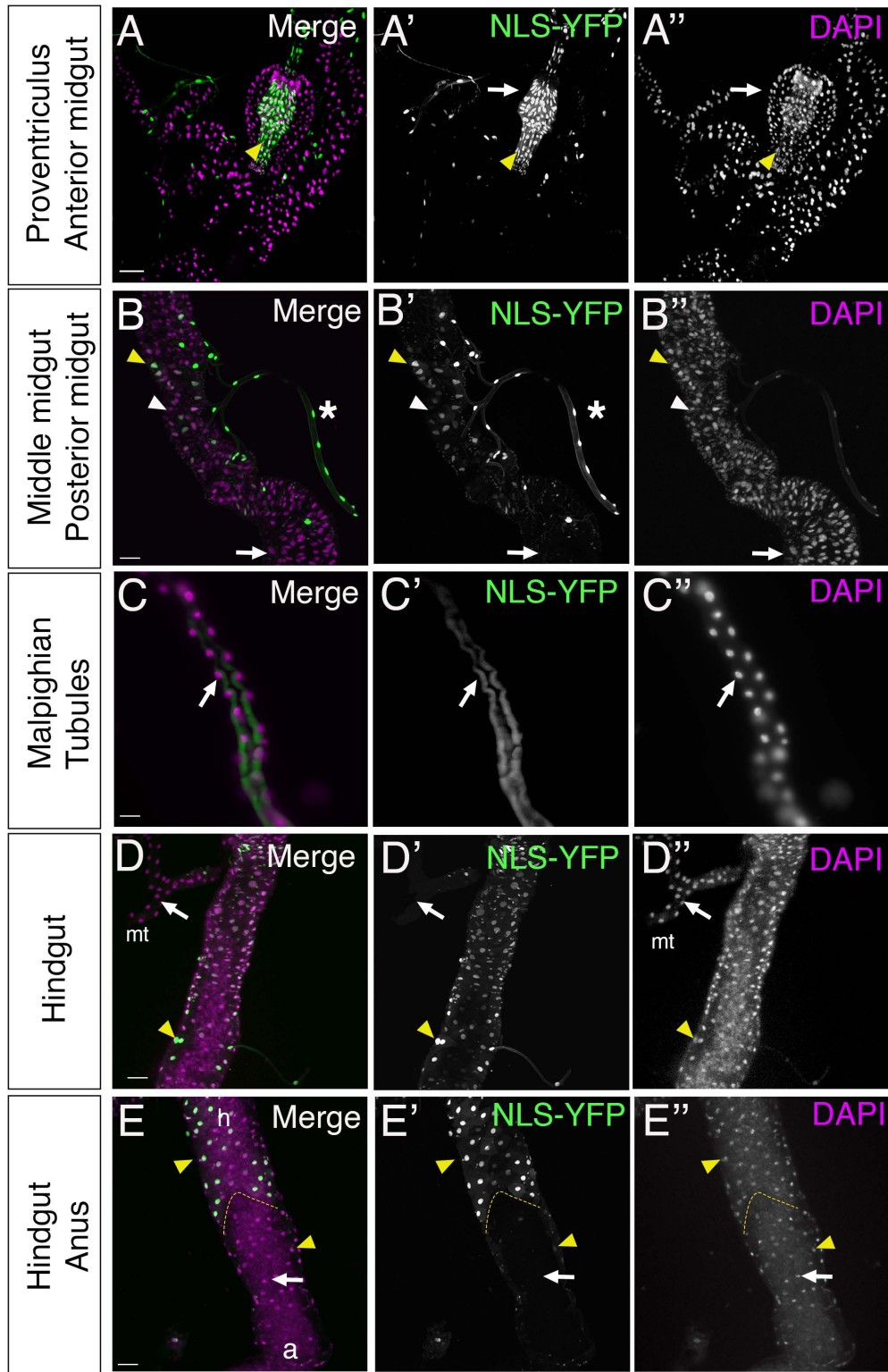

**Fig 2. miR-184 expression correlates with different class of septate junctions.** Intestines isolated from third instar larvae of *tubulin-LexA* crossed to *LexAOP-NLS-YFP* with YFP (green) with nucleus marker DAPI (magenta). All images are projections of multiple Z sections. (A) Proventriculus: NLS-YFP was absent from the peritrophic-membrane forming ring region and the cells anterior to it (white arrows). NLS-YFP was found colocalized with DAPI

in an area corresponding to foregut invagination and the wall of esophagus (yellow arrowhead) (6 larvae). (B) Middle midgut: NLS-YFP colocalized with DAPI in some nuclei in the middle midgut (yellow arrowhead) and was absent in others in this region (white arrowhead). Cells in the neighbouring trachea (white asterisks) were NLS-YFP positive. Cells in the posterior midgut (white arrow) flanking this region lacked NLS-YFP (6 larvae). (C) Malpighian tubules: NLS-YFP was absent in all the nuclei (white arrow) (6 larvae). (D) Hindgut: NLS-YFP was colocalized with DAPI (yellow arrowhead) in all nuclei. The flanking Malpighian tubules (mt) lacked YFP expression (white arrows) (6 larvae). (E) Hindgut-anus marginal zone: highlighted area shows the marginal zone between hindgut (h) and anus (a). Anus region had nuclei without (white arrow) and with (yellow arrowhead) NLS-YFP (6 larvae). Scale bars = 50 µm.

majority of the midgut expresses miR-184, except a subset of cells within the middle region, which may correspond to copper cells [44]. In Malpighian tubules, none of the nuclei expressed NLS-YFP (white arrows, Fig 2C, 2D). Thus, in tissues with sSJs, the midgut and Malpighian tubules, the majority of cells did not express NLS-YFP. We next analyzed the hindgut, which contains pSJs, and found that all nuclei in this region were positive for NLS-YFP expression (yellow arrowhead, Fig 2D). However, NLS-YFP expression was mostly absent at the terminus of the hindgut, in the anterior anal region, (white arrows, Fig 2E) with some cells weak NLS-YFP expression around the periphery (yellow arrowhead, Fig 2E). Overall, we found that the expression of NLS-YFP (the lack of miR-184) correlated with the presence of pSJ in tissues and those tissues with sSJ lacked NLS-YFP (express miR-184).

We next analyzed the expression of miR-184 in another tissue with pSJs, glia of the nervous system. miR-184 is expressed predominantly in the central nervous system of embryos and in the brain, eye discs and wing imaginal discs in larval stages [45]. To characterize the expression of miR-184 in glia and to differentiate between neuronal or glial cells, we used an antibody to Reversed polarity (Repo), a transcription factor that is specific to glial nuclei [46]. We observed robust NLS-YFP expression across a subset of glial cells in the brain lobes with little expression in neuronal populations (Fig 3A). Of interest was the absence of YFP expression in the glia of the eye disc (ed), which were strongly labeled with repo but lacked YFP expression (white arrow, Fig 3A). Within the brain lobe, we observed two clusters of neurons (non-Repo nuclei) that expressed NLS-YFP in the dorsal region of the brain lobe. An example of this cluster of neurons is shown in Fig 3D (yellow asterisks). Within the more superficial layers of the CNS brain lobe (Fig 3B), we found that the majority of Repo-positive nuclei were YFP positive (yellow arrowhead) with only a few exceptions (white arrow), along with YFP positive neurons (white asterisk, Fig 3B). There are a number of glial populations within this region, both perineurial glia and SPG along with the nuclei of the cortex glia. In contrast to the brain lobe, the expression of the NLS-YFP sensor in the ventral nerve cord (VNC) differed (Fig 3C) with most YFP expressing cells lacking Repo (white asterisk, Fig 3C) and a few co-labelled with Repo (yellow arrowhead, Fig 3C), suggesting miR-184 is not expressed in a large subset of neurons within the VNC.

Given the expression of the NLS-YFP in intestinal cells that contain pleated septate junctions we analyzed the distribution of the sensor in the subperineurial glia marker with lacZ using a Gli-lacZ marker [24]. LacZ is expressed in a subset of glia that mark the surface glial layers of the brain lobe and the ventral nerve cord (Fig 3D). Higher resolution revealed that NLS-YFP is observed in the lacZ positive SPG in the brain lobe (yellow arrowhead, Fig 3E) and other superficial nuclei lacking lacZ (white arrows, Fig 3E). Similarly, in the VNC, NLS-YFP was observed in the SPG on the surface of the VNC and the channel glia that wrap the axons exiting the VNC at the midline (yellow arrowheads, Fig 3F), another class of glia known to express pSJ [47,48].

We then expanded our analysis to the glia that ensheathe peripheral nerves. Interestingly, in the peripheral nerves, the NLS-YFP sensor was present in all glial nuclei throughout each abdominal nerve (yellow arrowhead, Fig 3G), including the perineurial, subperineurial and wrapping glia. Our NLS-YFP sensor study revealed that all peripheral glia and the majority glia in the superficial layers of the brain lobes do not express miR-184. This population also include glia that create pSJs. Given our observations that most tissues that generate critical permeability barriers including trachea, salivary glands, body wall epithelia, do not express miR-184 this suggests that the presence of miR-184 maybe tightly controlled in these cells to ensure pSJ integrity.

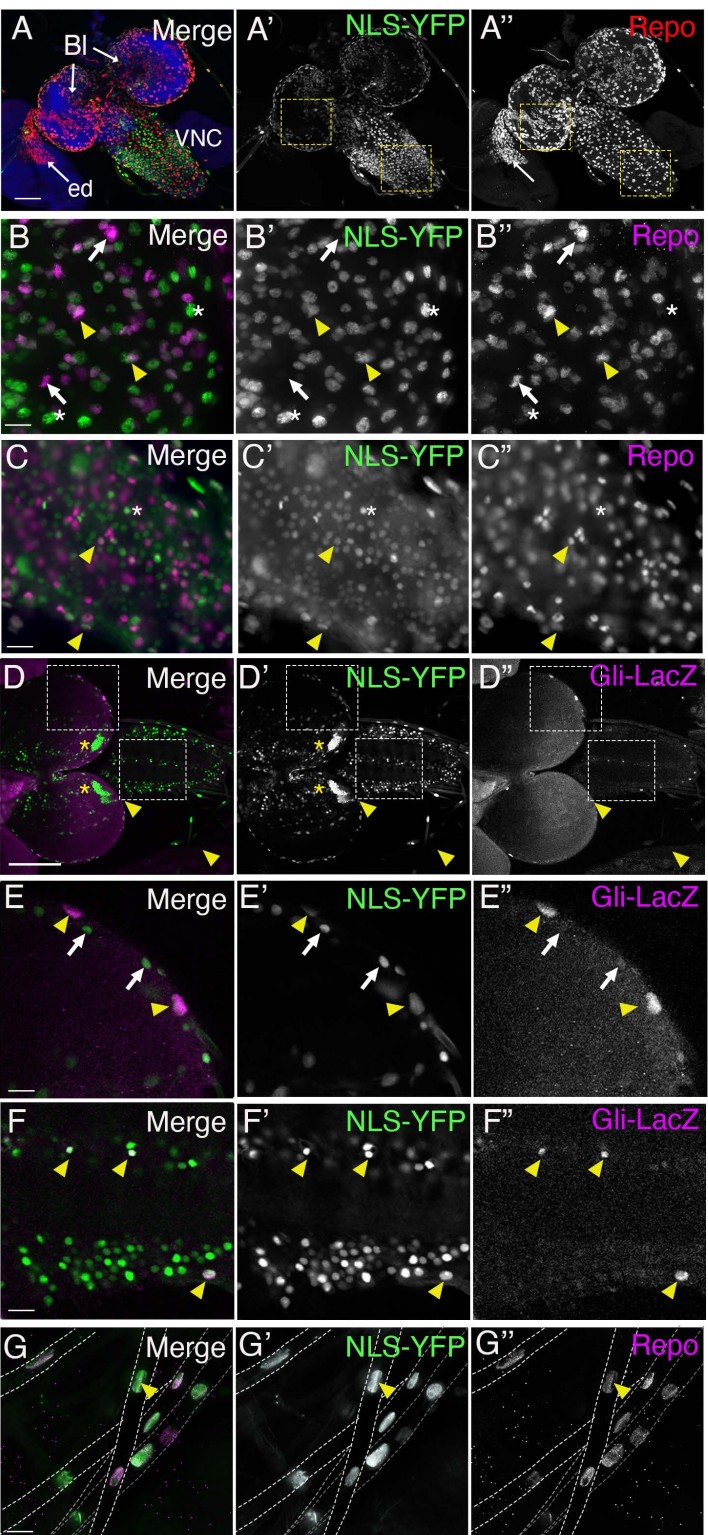

**Fig 3. miR-184 expression in the central and peripheral nervous system.** The nervous system from third-instar larvae of *tubulin-LexA* crossed to *LexAOP-NLS-YFP* with YFP (green). Panels represent z-projections of multiple z stacks. (A-D) YFP expression in comparison to glia immunolabelled with the glial transcription factor Repo (red, magenta). (6 larvae). (A) Overview of the nervous system showing the brain lobes (Bl), ventral nerve cord

(VNC) and eye disc (ed). YFP expression was observed throughout the brain lobes, ventral nerve cord and was absent from the eye disc (white arrow) (6 larvae). (B) Brain lobes: three different populations of nuclei were observed: Repo positive, NLS-YFP positive glial nuclei (yellow arrowheads), Repo positive, NLS-YFP negative glial nuclei (white arrows), Repo negative, NLS-YFP positive neuronal nuclei (white asterisks) (6 larvae). (C) Ventral nerve cord: More cells express NLS-YFP within the VNC including populations of glia (repo positive, yellow arrowheads) and neurons (white asterisk). (D-F) YFP expression (green) in comparison to nuclear localized lacZ (magenta) using the subperineurial glia marker, Gliotactin-lacZ. (3 larvae). (D) Lower resolution of CNS, with clusters of neurons in the brain lobe (yellow asterisks). Dashed boxes were digitally magnified 200% in E and F respectively. (E) Brain lobe: YFP expression was observed in the large nuclei of the subperineurial glia (lacZ, yellow arrowheads) along with other superficial cells (white arrowheads), possibly perineurial and cortex glia. (F) Ventral nerve cord: YFP expression was observed in the pSJ positive channel glia (top yellow arrowhead) and the subperineurial glia (lower yellow arrowhead) on the edges of the ventral cord. (G) Glia in peripheral nerves: peripheral nerve tracts are indicated by the dashed white lines. All Repo-positive glial nuclei throughout each peripheral nerve were positive for NLS-YFP (yellow arrowhead) (60 nerves in 6 larvae). Scale bars for (A, D): 50 µm. Scale bars for (B-C, E-G): 15 µm.

## Glial pSJs in the brain lobes are disrupted by miR-184 expression

Given the lack of expression of miR-184 in multiple barrier-forming tissues, we wanted to test and understand the consequence of miR-184 expression in these tissues. For this, we focused on the SPG, a class of glia that create the blood-brain and blood-nerve barriers, to study the role of miR-184 on pSJ regulation. As the composition of the pSJs and TCJ of the SPG were not fully characterized, our first step was to determine which of the potential miR-184 targets proteins were present in SPG pSJ and TCJ. miR-184 is predicted to target a suite of pSJ and TCJ mRNAs, many of which were verified *in vitro* [25]. We therefore focused our investigation on the predicted target pSJ proteins Nrx-IV, Kune, Sinu and Mcr, and the TCJ proteins, Gli and M6. We assessed distribution in both the brain lobes of the CNS and peripheral nerves of the PNS. Nrx-IV was used as a control and benchmark, given that its presence and barrier function are well characterized in both central and peripheral SPG [3,9].

To visualize the pSJs in the brain lobes, we used an *SPG-Gal4* line to drive the expression of a membrane-tagged RFP (mCD8::RFP) (*SPG>mCD8::RFP*) along with Nrx-IV endogenously tagged with GFP (Nrx-IV::GFP). pSJs in the outermost layer of the brain lobes form continuous thin lines outlining the point of contact of membranes of neighbouring SPG (arrows, Fig 4A–C). Using immunolabeling, we analyzed the distribution of Mcr, and the claudin-like proteins Kune and Sinu. Kune expression colocalized with Nrx-IV throughout the pSJ (arrows, Fig 4A–A"). Interestingly, Sinu expression was considerably weaker and only occasionally colocalized with Nrx-IV along the pSJ (yellow arrows, Fig 4BB"), rather immunolabeling appeared more dispersed within the SPG (white arrows, Fig 4B, B"). Thus, the expression of Kune was stronger in the brain lobe SPG and specific to the pSJ compared to Sinu. Similar to Kune, Mcr expression colocalized with Nrx-IV throughout the pSJs (yellow arrows, Fig 4C–C"). Overall, we found that in the brain lobes, Kune and Mcr consistently colocalized with Nrx-IV, along the length of the pSJs, while Sinu was mostly absent.

After confirming that the pSJ miR-184 target proteins are present in brain lobes, we checked if their expression pattern was affected by miR-184 overexpression. We utilized the *SPG-Gal4* driver with *Nrx-IV::GFP* and crossed this line to *UAS-miR-184* (*Nrx-IV::GFP, SPG-Gal4>UAS-miR-184*) and compared to our control cross to *w1118* (*Nrx-IV::GFP, SPG-Gal4*). In *UAS-miR-184* larvae, the pSJ structure was consistently disrupted. Two phenotypes were observed: 1. A moderate pSJ defect with intermittent interruptions in the structure (yellow arrowhead) with protein remnants along the pSJs (yellow asterisks, Fig 4E', F'). The pSJ was not continuous, but the overall pSJ structure could still be visualized by following the remnants; 2. A strong pSJ defect with pSJ protein aggregates (white asterisks) and some pSJ remnants (yellow arrowhead) (Fig 4D'), where the overall pSJ structure could not be traced. To score the defects, we first analyzed the Nrx-IV expression patterns. In *w1118* controls, 11% of the brain lobes exhibited a moderate phenotype (18 larvae), whereas in *UAS-miR-184* larvae, 100% of the brain lobes (28 larvae) exhibited moderate or strong phenotypes (Fig 4G). In all cases analyzed, the distribution of both Kune and Mcr colocalized with the Nrx-IV remnants (yellow arrowheads, Fig 4D", F"). The weak expression of Sinu did not allow us to make any conclusions through there was occasional localization of Sinu with the Nrx-IV remnants (yellow arrowhead, Fig 4E–E"). When quantified, 100% of the brain lobes expressing miR-184,

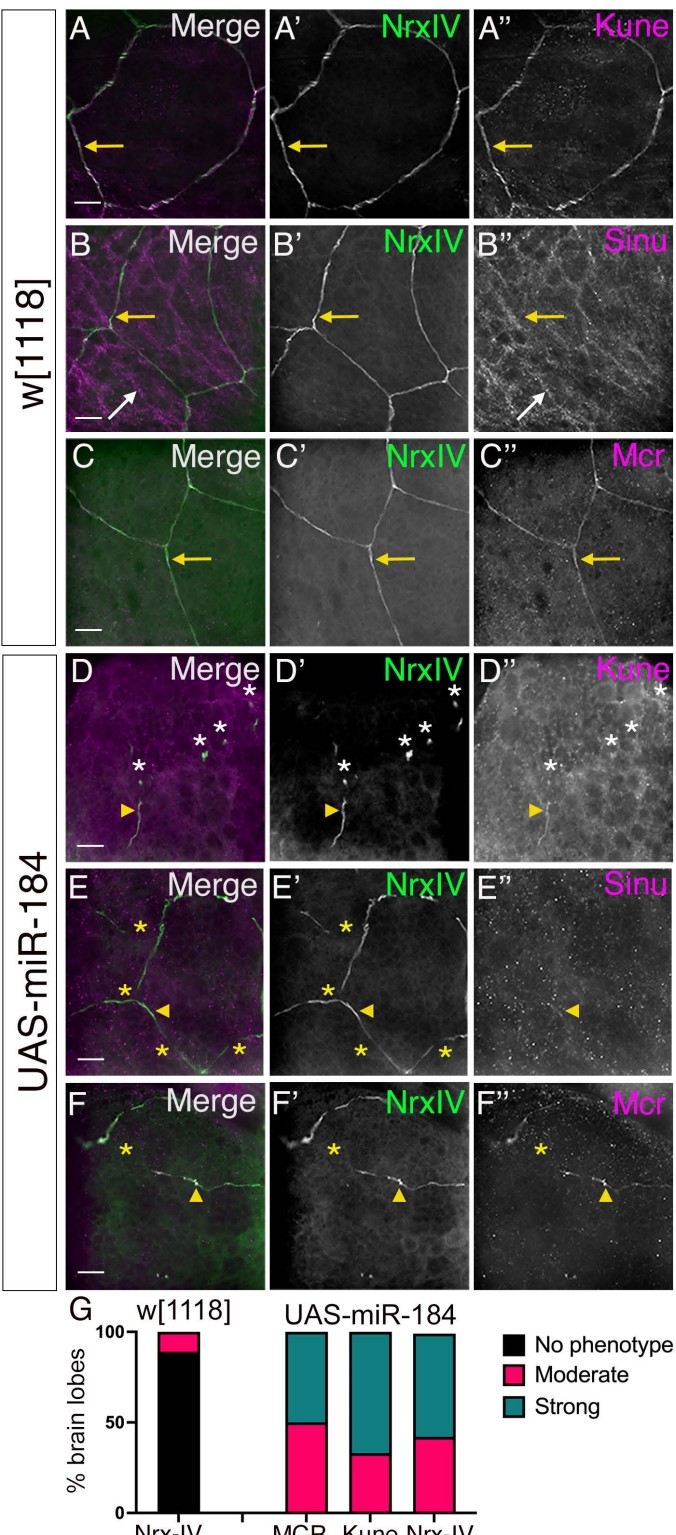

**Fig 4. Overexpression of miR-184 disrupts pleated septate junctions in the brain lobes.** Z-projection images of *SPG-Gal4* with Nrx-IV endogenously tagged with GFP (Nrx-IV::GFP; green) crossed to (A–C) control *w[1118]* and (D-F) *UAS-miR-184* and immunolabeled for Kune, Sinu and Mcr (magenta). (A) In controls, Kune colocalizes with Nrx-IV at pSJs (yellow arrows). (B) In controls, Sinu colocalizes with Nrx-IV at pSJs (yellow arrows) in

some areas and is expressed diffusely in SPG (white arrows). (C) In controls, Mcr colocalizes with Nrx-IV at pSJs (yellow arrows). (D) *UAS-miR-184*: a representative panel of a strong pSJ defect with only small sections of pSJ remaining (yellow arrowhead) and the formation of aggregates containing both Nrx-IV and Kune (white asterisks). (E) *UAS-miR-184*: a representative panel of a moderate pSJ defect with multiple gaps (yellow asterisks) in the pSJs. Weak Sinu expression (yellow arrowhead) corresponded to only some regions of Nrx-IV, with less expression in SPG compared to the control. (F) *UAS-miR-184*: representative moderate pSJ defect with multiple gaps (yellow asterisks) in the pSJ. Mcr colocalized with the Nrx-IV pattern (yellow arrowhead) in the remaining pSJs. (G) Quantification and characterization of the pSJ phenotypes in control (*w¹¹¹⁸*) and experimental (*miR-184*) brain lobes. Categorization of phenotypes: no phenotype (black), moderate (red) and strong (green). For *w[1118]*, Nrx-IV data is shown (n = 18). For *UAS-miR-184* larvae, all brain lobes displayed a phenotype (MCR n = 16; Kune n = 12; Nrx-IV n = 28). Scale bars: 15 µm.

Kune (12 larvae) and Mcr (16 larvae) displayed phenotypes similar to Nrx-IV (asterisks, Fig 4D–D", F–F"). Overall, we found that overexpression of miR-184 in the SPG led to interruptions in the pSJ resulting in remnants of the pSJ marked by Nrx-IV, Kune and Mcr.

### Glial pSJ proteins are disrupted by miR-184 in peripheral nerves

Next, we investigated the pSJ proteins within the peripheral nerves and the effect of miR-184 using the same lines as outlined above. In the peripheral nerves, SPG form autotypic pSJ junctions and bicellular pSJ junctions with the adjacent SPG cell [3]. When visualized using *Nrx-IV::GFP*, the pSJs formed a continuous thin line along the length of the SPG and at the points of contact between two neighboring SPG (yellow arrowheads, Fig 5A–C). Of the two claudin-like proteins, we found that Kune was expressed along the pSJs and colocalized with Nrx-IV expression (yellow arrowheads, Fig 5A–A"). Unlike in the brain lobes, Sinu was robustly expressed along peripheral nerve pSJs and colocalized with Nrx-IV (yellow arrowheads, Fig 5B–B"). Of note, we observed that Kune levels were higher brain lobe pSJs. To confirm this observation, we measured the normalized fluorescence intensity of Kune and compared these to Nrx-IV levels in the brain lobe versus peripheral nerves. The Kune to Nrx-IV ratio was significantly higher in the brain lobes compared to the peripheral nerves (p = 0.0052) (Fig 5H). Thus, Kune appears to be expressed at higher levels in the brain lobe compared to the peripheral nerves, while the opposite appears to be true for Sinu. Mcr was also expressed along the peripheral pSJs and colocalized with Nrx-IV (yellow arrowheads, Fig 5C) but displayed no differences between the CNS and PNS.

After confirming that the predicted miR-184 target proteins are in the peripheral pSJ, we tested the effect of miR-184 overexpression. Similar to the CNS, overexpression of miR-184 using the *SPG-Gal4* driver, disrupted the *Nrx-IV::GFP* pattern leading to loss of SJ proteins, breaks in the pSJ (yellow asterisks, Fig 5D–F') and protein remnants (white arrowhead, Fig 5D–F'). When quantified, in *w¹¹¹⁸* larvae 0% of nerves (351 nerves in 30 larvae) exhibited any pSJ defects, compared to *UAS-miR-184* larvae, where on average 85% of nerves (369 nerves in 32 larvae) exhibited pSJ defects (Fig 5G). Similar to Nrx-IV, Kune distribution was disrupted in all the larvae with 85% of the nerves affected (60 nerves in 6 larvae) compared to 0% in control (81 nerves in 6 larvae) (Fig 5G). Of note, the Kune immunolabeling colocalized with the Nrx-IV remnants (white arrowheads, Fig 5D–D") with similar interruptions in expression along the pSJs (yellow asterisks, Fig 5D–D"). In contrast, Sinu was completely absent from the Nrx-IV remnants (arrowheads, Fig 5E–E") and distribution was disrupted in 91% of nerves on average (89 nerves in 7 larvae) compared to 0% of controls (61 nerves in 5 larvae) (Fig 5G). Mcr protein also colocalized with the Nrx-IV remnants with breaks in its expression pattern along the pSJs (yellow asterisks, Fig 5F–F"). Mcr was disrupted in 65% of the nerves analyzed (67 nerves in 7 larvae) compared to 0% (75 nerves in 7 larvae) in the control (Fig 5G). Overall, we found that expression of miR-184 led to disruption of pSJ in both the brain lobe and peripheral nerves. Nrx-IV, Kune, and Mcr remained closely associated with what appears to be pSJ remnants in peripheral nerves similar to those in the brain lobes. In contrast, Sinu was only weakly in the brain lobes and completely lost from the Nrx-IV labelled remnants in peripheral nerves, suggesting Sinu has different protein stability or association at the pSJ.

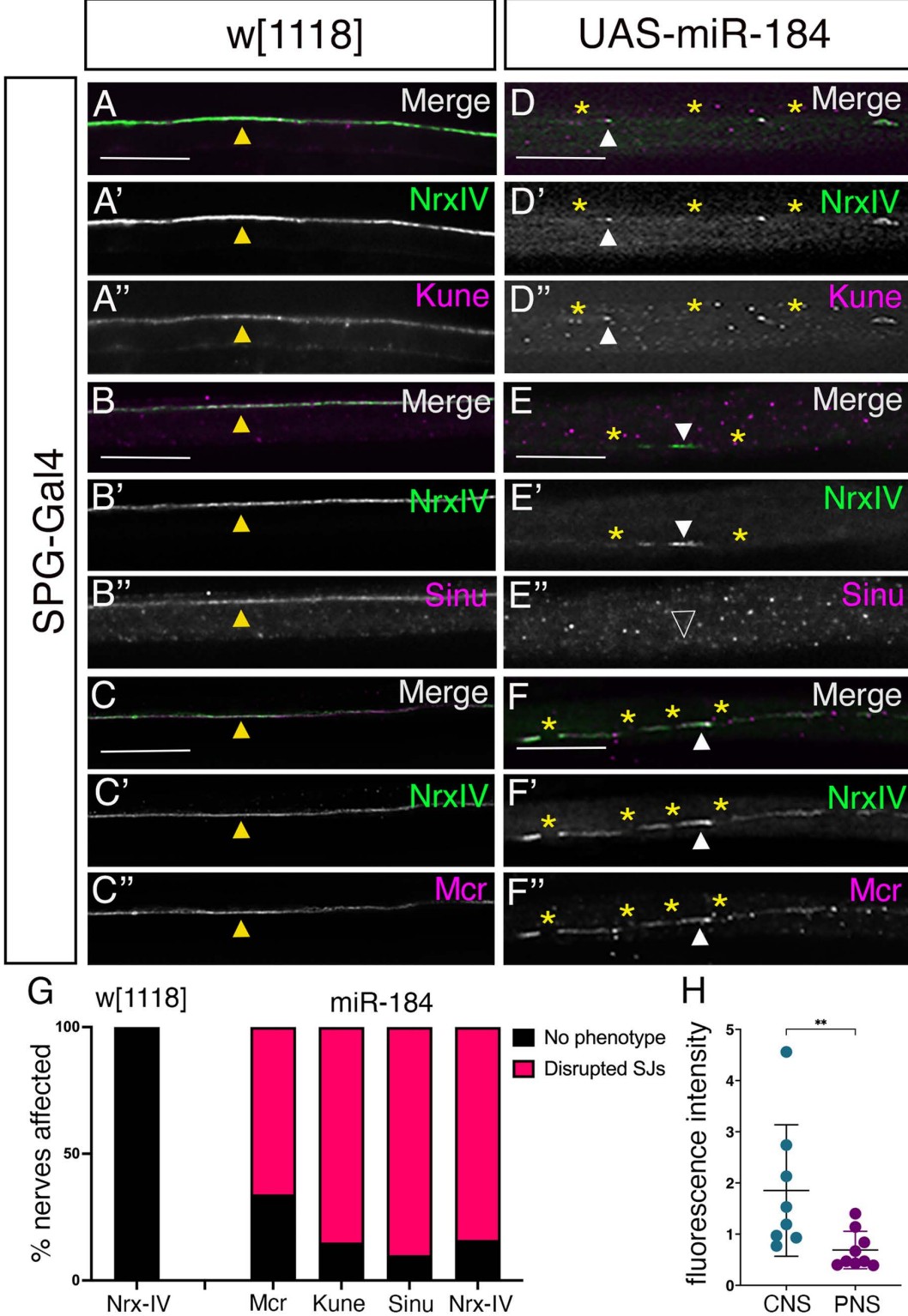

**Fig 5. Overexpression of miR-184 disrupts pleated septate junctions in peripheral nerves.** Z-projection images of *SPG-Gal4* with Nrx-IV::GFP (green) crossed to control *w[1118]* (A-C) and *UAS-miR-184* (D-F) and immunolabeled for Kune, Sinu and Mcr (magenta). (A) In controls, Kune colocalizes with Nrx-IV expression (yellow arrowhead) at pSJs (81 nerves in 6 larvae). (B) In controls, Sinu colocalizes with Nrx-IV expression (yellow arrowhead)

at pSJs (60 nerves in 6 larvae). (C) In controls, Mcr colocalizes with Nrx-IV expression (yellow arrowhead) at pSJs (75 nerves in 7 larvae). (D) *UAS-miR-184*: Interruptions (yellow asterisks) in the pSJ with Nrx-IV remnants (D'), which colocalize with Kune remnants (D", white arrowheads) (60 nerves in 6 larvae). (E) *UAS-miR-184*: Interruptions (yellow asterisks) in the pSJ with Nrx-IV remnants (E', white arrowhead), with no Sinu expression (E", empty arrowhead) (89 nerves in 7 larvae). (F) *UAS-miR-184*: Interruptions (yellow asterisks) in the pSJ with Nrx-IV remnants (F'), which colocalize with Mcr remnants (F", white arrowheads) (67 nerves in 7 larvae). (G) Quantification and characterization of the pSJ phenotypes in control (*w[1118]*) and experimental (*miR-184*) larvae for Nrx-IV, Kune, Sinu and Mcr. Categorization of phenotypes: no phenotype (black), phenotype (red). For *w[1118]*, Nrx-IV data is shown (n = 30 larvae). For *UAS-miR-184* larvae, the majority of nerves had disrupted pSJs (Mcr n = 7 larvae; Kune n = 6 larvae; Sinu n = 7 larvae; Nrx-IV n = 32 larvae). (H) Comparison of Kune mean fluorescence intensity (M.F.I) normalized to Nrx-IV levels in the brain lobes (8 larvae) and peripheral nerves (9 larvae). Statistical significance determined by Mann-Whitney U test (p = 0.0052). The mean plus standard deviation is indicated, and each data point is a single larva. Scale bars: 15 µm.

## TCJ proteins are not affected by miR-184

After establishing which pSJ proteins are present and regulated by miR-184, we investigated the predicted miR-184-targeted TCJ proteins M6 and Gliotactin (Gli). To identify the TCJ, we analyzed Bark (aka Anakonda), a TCJ protein that is not a predicted miR-184 target. Bark is essential for the localization of Gli and is mutually dependent on M6 for localization to the epithelial TCJ [19,31,49]. We used *SPG-Gal4* and Bark endogenously tagged with GFP (*Bark::GFP)* crossed to *w^1118* and immunolabeled with Kune and Gli antibodies to visualize the pSJ and TCJ proteins, respectively. In the *w^1118* brain lobes, the point of contact of three pSJ from neighbouring SPG was visualized using Kune antibody and the TCJ with Bark (yellow arrowhead, Fig 6A–B'''). Gli colocalized with Bark and was concentrated at the TCJ (yellow arrowhead, Fig 6A–B''') and as puncta along the pSJ (white arrowhead, Fig 6A–B'''). With the *SPG-Gal4* driving expression of *UAS-miR-184*, the loss of Kune (yellow asterisks, Fig 6C–D''') made identification of the convergence of three pSJs difficult. However, Bark was retained as puncta presumably in the TCJ associated with Kune remnants (yellow arrowheads, Fig 6C–D'''). Gli was always observed associated with Bark puncta in the presumptive TCJ areas (yellow arrowhead, Fig 6C–D''') and at bicellular puncta (white arrowhead, Fig 6C–D''') that colocalized with Kune remnants. Thus, both Gli and Bark localize to the TCJ in the brain lobe, and this distribution appears to be strongly linked with pSJs in control and remnants of pSJs with miR-184 expression.

In the peripheral nerves of *SPG-Gal4; Bark::GFP* crossed to *w^1118*, the TCJ proteins Bark and Gli colocalized at the point of contact of three SPG membranes marked with the pSJ marker, Kune (yellow arrowhead, Fig 6E–E'''). As with the SPG in the CNS, the TCJ proteins were intermittently expressed along the pSJ as puncta (yellow arrowheads, Fig 6E–E''') in all the nerves analyzed. With overexpression of miR-184, the presumptive TCJ area in the peripheral nerves could not be determined, given the disruption of the pSJ morphology observed with Kune expression (yellow asterisks, Fig 6F–F'''). However, the expression of Gli and Bark colocalized with each other and with remnants of Kune in all nerves analyzed (yellow arrowhead, Fig 6F–F'''). Thus, Bark and Gli were present and localized to the Kune remnants with miR-184 overexpression in SPG in the brain lobes and peripheral nerves.

We next tested for another TCJ protein M6, which is key to the formation of the TCJ in epithelia and predicted to be targeted by miR-184. Using Nrx-IV::GFP to determine the point of contact of three pSJs and the TCJ, immunolabeled M6 was present at the TCJ of brain lobe SPG (yellow arrowheads, Fig 7A–B"). M6 formed puncta at the TCJ (yellow arrowheads, Fig 7A–B") and was distributed laterally along the bi-cellular pSJs at small concentrations or puncta (white arrowheads, Fig 7A–B"). Thus, M6 is part of the TCJ complex in the brain lobes. To investigate the impact of miR-184 overexpression on M6 localization, we crossed *SPG-Gal4, Nrx-IV::GFP* to *UAS-miR-184* and found M6 protein was localized to the putative TCJ with Nrx-IV with both moderate and strong pSJ disruptions (yellow arrowhead, Fig 7C–D"). To determine the presence of M6 protein in peripheral nerves, we used the *SPG-Gal4* crossed to *w^1118*. *Nrx-IV::GFP* expression labeled the pSJs (white arrowhead; Fig 7E–E") and at the point of contact of three SPG membranes at the TCJ (yellow arrowhead, Fig 7E–E"). M6 was concentrated at the TCJ and also in the same puncta along the pSJ (white arrowhead, Fig 7E–E"). In the *UAS-miR-184* larvae, the loss of Nrx-IV and the pSJ structure made it hard to visualize

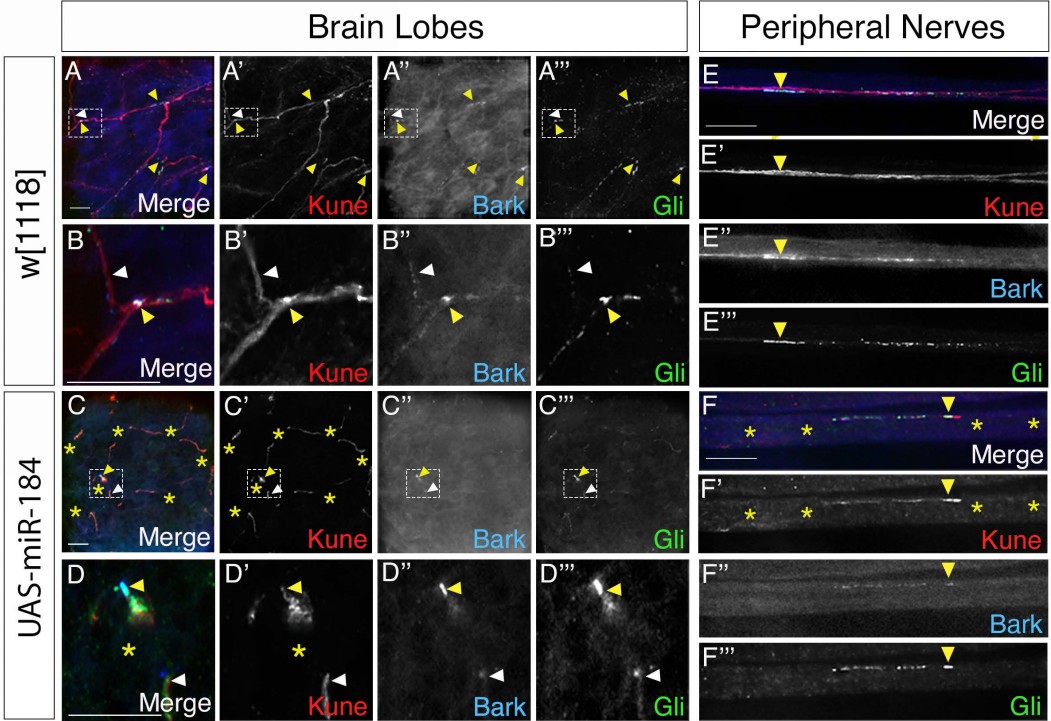

**Fig 6. TCJ protein localization in brain lobes and peripheral nerves.** *SPG-Gal4* with *Bark::GFP* (blue) crossed to control, *w[1118]* and to *UAS-miR-184* immunolabelled for Kune (red) and Gli (green). (A) In control brain lobes, pSJs visualized by Kune, with Bark and Gli expressed strongly at TCJ (yellow arrowhead) and as puncta along the pSJ (white arrowhead) of brain lobes. Highlighted TCJ area was magnified 500X visualized in (B) (10 larvae). (C) In *UAS-miR-184* brain lobes, disrupted pSJs were visualized by Kune (yellow asterisks) with Bark and Gli colocalization along the remaining TCJ (yellow arrowhead). Highlighted TCJ area was magnified 500X and visualized in (D) (8 larvae). (E) In control peripheral nerves, Kune labelled pSJs with Bark and Gli colocalized at the TCJ (yellow arrowhead) and as puncta along the pSJs (white arrowhead) (106 nerves in 10 larvae). (F) In *UAS-miR-184* peripheral nerves, disrupted pSJs visualized by Kune expression (yellow asterisks) with Bark and Gli colocalized with each other and along Kune remnants (yellow arrowhead) (77 nerves in 8 larvae). Scale bars: 15 μm.

the accurate location of the TCJ expression in the peripheral nerves (white asterisks, Fig 7F–F"). M6 was still present in areas without Nrx-IV expression (white arrowhead, Fig 7F, F") and often colocalized with the Nrx-IV remnants (yellow arrowhead, Fig 7F, F").

Overall, we found M6 and Gli were expressed in the TCJ and along sections of the pSJ in both brain lobes and peripheral nerves. Although both M6 and Gli mRNAs are predicted targets of miR-184 (targetscanfly.org), and the long mRNA isoform of Gli has been experimentally validated as a target in the wing imaginal disc [26], our studies showed that the overexpression of miR-184 in the SPG did not result in a detectable reduction of either protein at the TCJ of glial cells. Additionally, we found that the TCJ proteins are still associated with remnants of the pSJ, suggesting the complex formed at the convergence of pSJ at the TCJ is relatively stable in both the brain lobe and the peripheral SPG.

### miR-184 expression compromises the blood-brain-barrier and locomotion

After confirming that miR-184 overexpression in SPG affects the pSJ protein expression pattern, we investigated whether there were also defects in the permeability barrier function. To test the blood-brain barrier integrity in third-instar larvae, we performed a dye penetration assay with a 10,000 MW Dextran conjugated to Texas Red dye as previously described

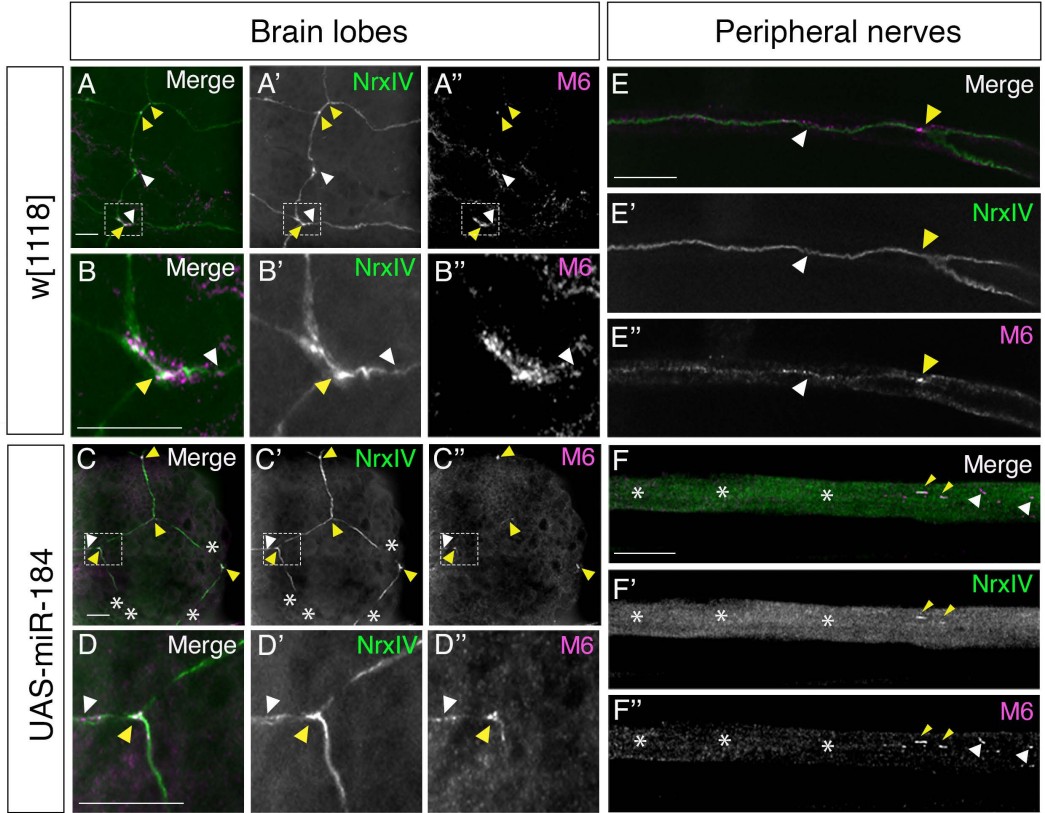

**Fig 7. M6 localization in brain lobes and peripheral nerves.** (A–F) *SPG-Gal4; Nrx-IV::GFP* crossed to *w¹¹¹⁸* (control) and, to *UAS-miR-184*, immuno-labeled for M6 (magenta) with Nrx-IV (green) as a pSJ marker. (A) In control brain lobes, pSJs visualized by Nrx-IV with M6 expression at TCJ (yellow arrowhead) and puncta distributed along pSJs (white arrowhead). Highlighted TCJ area was magnified 500X and visualized in (B) (6 larvae). (C) In *UAS-miR-184* brain lobes, moderate pSJ defects visualized by Nrx-IV (white asterisks) with M6 expression along the TCJ remnants (yellow arrowhead) and puncta along the pSJs (white arrowhead). Highlighted TCJ area was magnified 500X and visualized in (D) (6 larvae). (E) In control nerves, pSJs visualized by Nrx-IV with M6 expression at the TCJ (yellow arrowhead) and as puncta along pSJs (white arrowhead) (67 nerves in 6 larvae). (F) In *UAS-miR-184* nerves, disrupted pSJs observed by Nrx-IV (white asterisks) with M6 expression colocalized with Nrx-IV remnants (yellow arrowhead) and expressed alone in some regions (white arrowhead) (81 nerves in 6 larvae). Scale bars: 15 µm.

[39]. For this, we labeled SPG membranes using Gli-Gal4 driving a membrane-tagged GFP (*Gli-Gal4>mCD8::GFP*) and crossed this line to *UAS-miR-184*, and to the negative and positive controls, *UAS-lacZ* and *UAS-Nrx-IV-RNAi,* respectively (Fig 8A–C). In the negative control (*Gli>lacZ*) larvae, the dye did not enter the brain lobes or ventral nerve cord (Fig 8A). While in the positive control (*Gli>Nrx-IV-RNAi)*, dye readily permeated the brain and ventral nerve cord (Fig 8C). When quantified in the neuropile region of the brain lobes, the fluorescence intensity in the Nrx-IV-RNAi larvae was significantly greater than in the lacZ larvae (p=0.0472) (Fig 8D). Similarly, in the *Gli>miR-184* larvae, the dye permeated to the brain and ventral nerve cord (Fig 8B), and when quantified the fluorescent intensity was significantly greater than the *lacZ* control (p<0.001) (Fig 8D). Thus, we found that overexpression of miR-184 in the SPG compromises the pSJ and the blood-brain barrier.

After confirming that the blood-brain barrier was compromised, we next investigated if this would affect neuronal functions. Previous research has shown that mutations in core pSJ proteins, essential for barrier formation, disrupt the neuronal signaling environment and result in impaired mobility [50,51]. Therefore, we hypothesized that a compromised blood-brain barrier might affect larval locomotion. We conducted larval locomotion assay [36] using *SPG-GAL4* crossed to

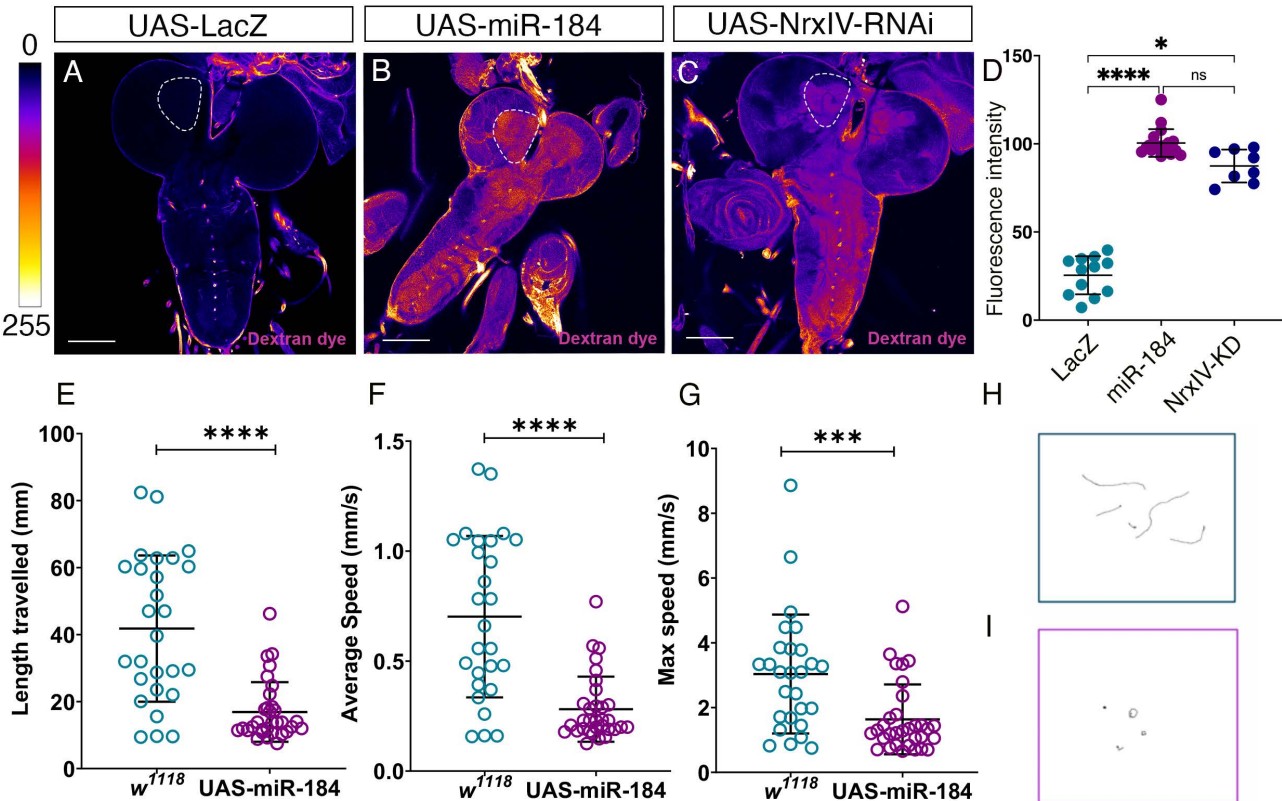

**Fig 8. Overexpression of miR-184 disrupts the blood-brain barrier and larval locomotion.** (A–D) miR-184 expression compromises the blood-brain barrier. (A-C) *Gli-Gal4* driven in SPG was crossed to the controls (A) *UAS-LacZ* (12 larvae), (B) *UAS-Nrx-IV-RNAi* (8 larvae) and (C) *UAS-miR-184* (18 larvae). Highlighted area in white shows the area where the Texas Red dye permeability was measured. Degree of the fluorescence is indicated using the Fire LUT (0-255) with more dye permeation indicated by lighter colors. Scale bars = 50 µm. (D) Mean Fluorescence Intensity (M.F.I) quantification of dye permeation in the neuropile region. The mean and standard deviation are shown, and each data point represents a single larva. Statistical significance determined by Kruskal-Wallis's test with Dunn's multiple comparisons test between groups (* p = 0.047; **** p < 0.001, ns not significant). (E-I) miR-184 expression disrupts larval locomotion. (E-G) Locomotion tracking of third instar larvae in *SPG-Gal4* crossed to *UAS-miR-184* (32 larvae) and to control *w1118* (27 larvae) with (E) length travelled in mm, (F) average speed in mm/s and (G) maximum speed in mm/s. The mean and standard deviations are shown. Statistical differences determined for each parameter using the Mann-Whitney test (**** p < 0.0001; *** p = 0.0004). (H-I) Representative path tracing image depicting larval movements of (H) *w[1118]* and (I) *UAS-miR-184* larvae recorded over 1 minute.

*w1118* as control or to *UAS-miR-184*. miR-184 overexpression led to a significant decrease in all three parameters of locomotion measured, length traveled (****p < 0.0001), average speed (****p < 0.0001), and maximum speed (***p = 0.0004) compared to control *w1118* (Fig 8E-G). The path tracking images clearly showed that *w1118* larvae traveled in a consistent motion and direction away from the center of the plate, whereas the *UAS-miR-184* larvae did not travel in any particular direction and had moved less distance (Fig 8H, I).

After determining that the blood-brain barrier and larval locomotion were severely affected when miR-184 was overexpressed in the SPG, we investigated whether they could survive post-third instar larval stages. We found that expression of miR-184 in the SPG led to 100% lethality at the late larval stage (32 larvae) when raised continuously at 25°C, compared to the control, where all the larvae (27 larvae) pupated and hatched. To confirm this effect was specific to the SPG, we used another SPG driver (Mdr65-Gal4) to drive the expression of miR-184 and also observed 100% lethality at the late larval stage. Thus, overexpression of miR-184 in the SPG leads to a compromised blood-brain barrier with locomotory defects and failure to survive beyond the larval stages.

## miR-184 overexpression does not alter target SJ or TCJ mRNA levels

After determining the consequences of miR-184 expression on pSJ and TCJ proteins, we investigated the potential mechanism of this regulation. Since miR-184 overexpression in SPG led to a reduction in the predicted pSJ target proteins, we next tested if mRNA levels were also affected. miR-184 is predicted to target the pSJ mRNAs Nrx-IV, Kune, Mcr and Sinu and the TCJ mRNAs Gli and M6 (targetscanfly.org) (S1 Fig). Therefore, we performed qRT-PCR analysis using mRNA isolated from the dissected nervous systems of third-instar larvae (Fig 9A). We tested mRNA levels of the pSJ proteins and for the TCJ proteins, Gli and M6. We also tested pSJ protein, Nervana2 (Nrv2), along with the TCJ protein, Bark, both of which are not predicted targets of miR-184. The quantification cycle values (Cq) for each primer/mRNA were measured and normalized with the corresponding values for actin to obtain the ΔCt value. For ΔΔCt value for each mRNA, the ΔCt control value (*Gli-Gal4 > UAS-LacZ*) was subtracted from the experimental value (*Gli-Gal4 > UAS-miR-184*). Interestingly, across the three biological replicates (representing each dot on the graph), the qRT-PCR results showed that overexpression of miR-184 had no impact on the mRNA levels of the pSJ proteins Nrx-IV, Mcr, Kune and Sinu and the TCJ proteins, Gli, M6 or Bark (Fig 9A). While the levels of Kune were reduced, the difference was not statistically significant compared to the other pSJ and TCJ mRNAs (Fig 9A). Surprisingly, there was an increase in the mRNA levels of the control Nrv2, and this increase was significant compared to the Kune, but not relative to the other pSJ and TCJ proteins (Fig 9A). Overall, miR-184 overexpression did not significantly affect the mRNA levels of any predicted targets but instead showed an unexpected increase in Nrv2 expression.

## miR-184 overexpression leads to increased expression and mislocalization of Nrv2

We then focused on Nrv2 further due to the increase in mRNA levels. We confirmed expression of Nrv2 protein in the peripheral nerve and characterized its expression pattern in the brain lobe. We used the same genotype as before, *SPG-Gal4; Nrx-IV::GFP* crossed to *w1118* and detected Nrv2 protein using a Nrv2.1 antibody. In control brain lobes, while the Nrv2 expression was weak, it did colocalize with Nrx-IV (yellow arrows, Fig 9B–B"). In the *UAS-miR-184* brain lobes, Nrv2 was localized to only a few regions that corresponded to Nrx-IV pSJ remnants (yellow arrowhead, Fig 9C–C"). However, owing to the weak expression pattern even in the controls, we could not draw any conclusions on the Nrv2 protein level increase in the brain lobes. Nrv2 is an established pSJ protein in peripheral nerves [3,9] and thus as expected in the *w1118* controls, Nrv2 strongly colocalized with Nrx-IV expression throughout the pSJs (yellow arrows, Fig 9D–D"). With overexpression of miR-184, we observed Nrv2 colocalized with the Nrx-IV remnants in some regions (yellow arrowhead, Fig 9E–E"). However, Nrv2 expression was not confined to the pSJ remnants but rather expression expanded and spread throughout the SPG membranes (yellow asterisks, Fig 9E"). Upon quantification of the fluorescent intensity of Nrv2 within the SPG, we found that Nrv2 levels were significantly higher in the *UAS-miR-184* larvae than in controls (Fig 9H). Thus, overexpression of miR-184 in SPG resulted in higher levels of Nrv2 protein and was found to spread throughout the SPG membrane. Overall, our findings show that overexpressing miR-184 in SPG not only led to defects in its pSJ targets but also led to a change in the levels and distribution of Nrv2.

Our results suggest that changes in localization of some pSJ protein (like Kune, Mcr, Sinu) can trigger compensatory changes in other pSJ mRNA and protein levels (like Nrv2). As miR-184 targeted multiple pSJ proteins, we next investigated if the Nrv2 mislocalization would also result from the loss of just one of the core SJ proteins, Nrx-IV. To downregulate Nrx-IV in SPG, we crossed another SPG driver, *Gli-Gal4,* to *UAS-Nrx-IV-RNAi*, with *UAS-LacZ* as the control. In *UAS-LacZ* larvae, Nrv2 expression was observed in the expected pattern along the pSJ (yellow arrows, Fig 9F–F'), while Gli was observed in the expected pattern at the TCJ structure (white arrowheads, Fig 9F") and as puncta along the pSJ. With the knockdown of Nrx-IV, we found that Nrv2 was mislocalized along the SPG membrane, similar to the miR-184 overexpression phenotype (yellow asterisks, Fig 9G–G'), and when quantified, the fluorescence intensity of Nrv2 was significantly higher than control (Fig 9I). Thus, the expression level of Nrv2 was increased when the Nrx-IV expression was reduced. Interestingly, the knockdown of Nrx-IV led to TCJ protein mislocalization as well. Gli protein expression was

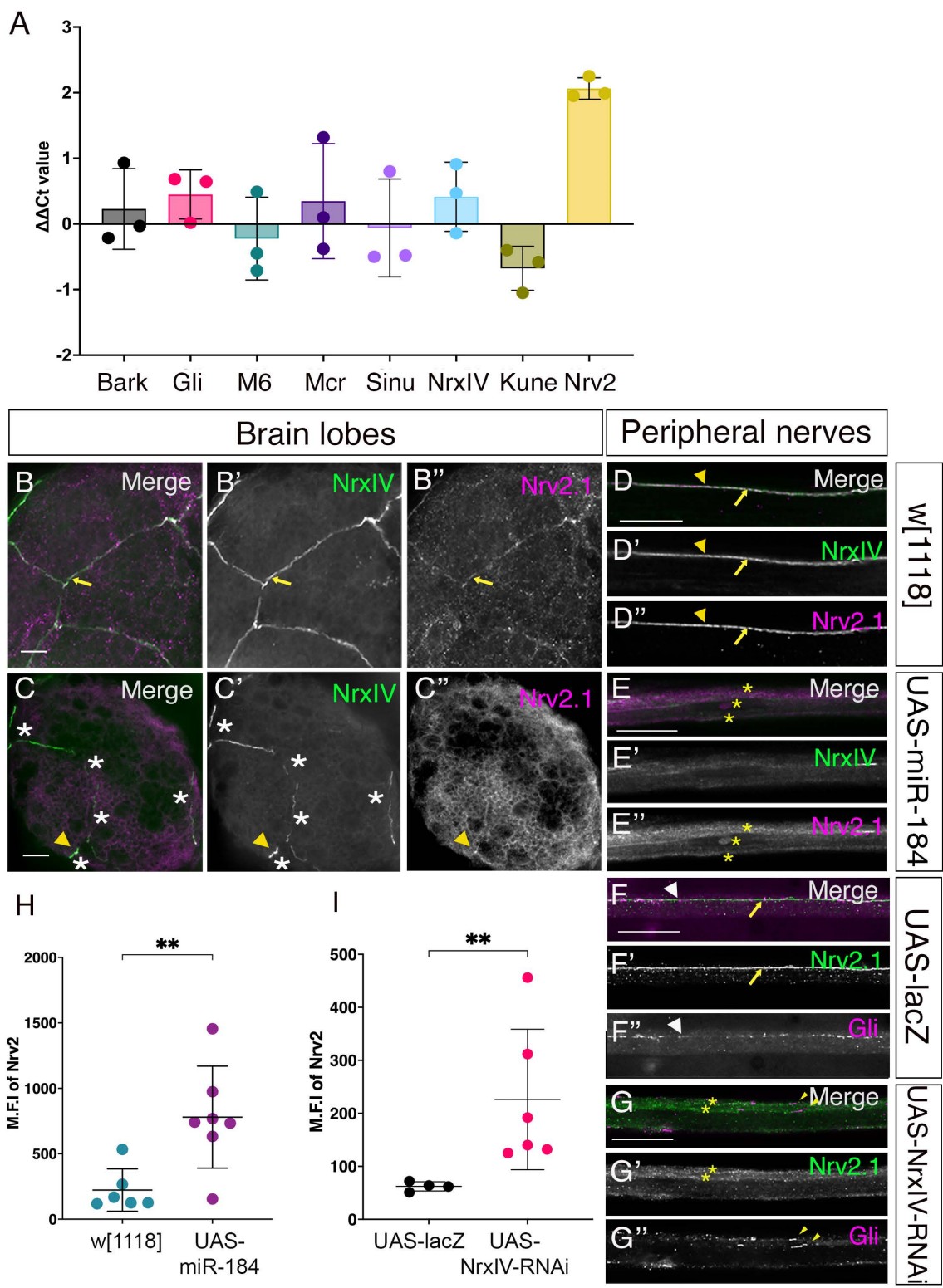

**Fig 9. miR-184 does not downregulate target pSJ and TCJ mRNA levels but changes Nrv2 expression.** (A) qRT-PCR of miR-184 targets mRNA. pSJ (Mcr, Sinu, Nrx-IV, Kune) and TCJ (Gli, M6) mRNAs compared to non-target mRNAs, Bark and Nrv2. *SPG-Gal4; Nrx-IV::GFP* crossed to *w^1118* (control) and *UAS-miR-184* (experimental). The mean ΔΔCt value was calculated by subtracting normalized quantification cycle values (Cq) of control

from the experimental genotype. The mean and standard deviation are shown with each data point representing a biological replicate of 20 larvae in each. Statistical differences were calculated using the Kruskal-Wallis and Dunn's multiple comparisons test between different groups (*p = 0.024, non-significant values are not shown). (B-I) Nrv2 is mislocalized with miR-184 overexpression and Nrx-IV knockdown. (B-E) *SPG-Gal4* with Nrx-IV endogenously tagged with GFP (green) crossed to the control *w1118* and *UAS-miR-184*, and immunolabeled for Nrv2.1 (magenta). (B) In control brain lobes, pSJs visualized by Nrx-IV, weakly colocalized with Nrv2 (yellow arrows) (5 larvae). (C) In *UAS-miR-184* brain lobes, disrupted pSJs (white asterisks) are visualized by Nrx-IV remnants (yellow arrowhead) in a representative image for a pSJ moderate defect, which weakly colocalized with Nrv2 expression (yellow arrowhead) (7 larvae). (D) In control peripheral nerves, pSJs visualized by Nrx-IV, which colocalized with Nrv2 (yellow arrows) (85 nerves in 6 larvae). (E) In *UAS-miR-184* peripheral nerves, Nrx-IV remnants (yellow arrowhead) colocalized with some Nrv2 remnants (yellow arrowhead). Nrv2 expression is mislocalized and spread around the membrane (yellow asterisks) (77 nerves in 7 larvae). (F-G) Peripheral nerves of *Gli-Gal4* crossed to the control, *UAS-LacZ* and to *UAS-Nrx-IV-RNAi* immunolabelled with Nrv2.1 (green) and Gli (magenta). (F) In control nerves, Nrv2 labelled the pSJs (yellow arrows) and Gli was expressed in puncta along the pSJs (white arrowheads) (59 nerves in 5 larvae). (G) In the *UAS-Nrx-IV-RNAi* nerves, Nrv2 (yellow asterisks) and Gli (yellow arrowheads) mislocalized and spread along the SPG membrane (69 nerves in 6 larvae). (H-I) Bar graphs quantifying the mean fluorescence intensity (M.F.I) of Nrv2. Each point represents average intensity of Nrv2 for one larva. Error bars represent the standard deviation from the mean. (H) Quantification of mean fluorescence intensity (M.F.I) of Nrv2 with miR-184 overexpression: Bar graphs for control *w1118* (green) and the *UAS-miR-184* (magenta). The M.F.I was higher with miR-184 overexpression compared to the control; statistical significance determined by Mann-Whitney U test (** p = 0.008). (I) Quantification of M.F.I of Nrv2 with Nrx-IV knockdown: Bar graphs for the control, *UAS-LacZ* (green) and the *UAS-Nrx-IV-RNAi* (magenta). The M.F.I was higher with miR-184 overexpression compared to the control; statistical significance determined by Mann-Whitney U test (** p = 0.0095). Scale bars: 15μm.

spread along the SPG membrane as two parallel lines of puncta along the pSJ (yellow arrowheads, Fig 9G, G"). However, because the presumptive TCJ could not be located due to mislocalized Nrv2 protein, no conclusions could be drawn about the localization of Gli to the TCJ. Nevertheless, we concluded that both Nrv2 and Gli are mislocalized with downregulation of Nrx-IV. Overall, these results indicate that expression of miR-184 affects pSJ morphology without degrading the mRNA, suggesting that other mechanisms, such as translational repression, might underlie the observed pSJ disruption.

## Discussion

Our primary goal was to investigate the distribution of miR-184 within the third-instar larvae and determine if miR-184 expression in glia can target and disrupt pSJs and permeability barrier formation. While the overall tissue distribution of miR-184 is not well understood, miR-184 is a maternally deposited microRNA [52] and is expressed in a highly dynamic pattern throughout fly development [45]. Using our NLS-YFP miR-184 sensor, we observed that the absence of miR-184 expression correlated with tissues that express pSJs and form critical permeability barriers including trachea, the sub-perineurial glia, and the hindgut. On the other hand, many tissues known to generate sSJs expressed miR-184 and these included the Malpighian tubules and the midgut, with the exception of the middle midgut, where a subset of cells (likely copper cells) did not express miR-184. It is possible that these tissues use miR-184 to ensure that pSJ proteins are not expressed. An exception to the correlation between tissues with pSJ and a lack of miR-184 expression were the imaginal discs. We found that, except for a few clusters, most cells of the wing imaginal discs expressed miR-184. Interestingly, our prior work found that increased miR-184 expression in the wing imaginal leads to the loss of pSJs proteins Nrx-IV, Mcr and the TCJ protein Gli without any deleterious effects such as loss of polarity [26]. Thus, the presence of miR-184 in this tissue would theoretically disrupt the pSJ structure. However, imaginal discs are highly proliferative epithelia undergoing extensive rounds of mitosis to expand in size, a process that requires continuous re-assembly of pSJs and TCJs [53]. In addition, miR-184 is known to regulate a range of signaling pathways including Saxophone (a DPP receptor), the transcriptional repressor Tramtrack69 (TTK69), and a gurken regulator K10 [52]. Thus, these dynamic cells might require baseline miR-184 expression to control levels of pSJ/TCJ proteins or other signaling pathways, compared to more static tissues with more stable pSJ structures found in glia and other permeability barrier tissues [54].

On the other hand, the absence of miR-184 expression strongly correlated with the presence of pSJs in tissues that are collectively polyploid and post-mitotic, including the hind gut, trachea, salivary glands, and subperineurial glia. These tissues express a similar composition of pSJ proteins [8,9,11–13,15]. Thus, it is possible that miR-184 is not expressed in

these tissues to ensure the integrity of the permeability barriers they create. Of interest is the observation that all classes of peripheral glia did not express miR-184 including the perineurial and wrapping glia, neither of which are known to express pSJ proteins. Perineurial glia contribute to the blood-brain barrier and wrapping/ensheathing glia can form internal barriers within the CNS [3]. Thus, is possible that miR-184 targets other subsets of proteins beyond the pSJ complex necessary for glial ensheathment and barrier formation.

To understand the consequences of expressing miR-184 in the polyploid tissues expressing pSJ proteins, we tested the effects in the blood-brain barrier/subperineurial glial cells and analyzed the pSJ structure. miR-184 expression led to the reduction in Nrx-IV and Mcr along with the claudin homologs Kune and Sinu, matching our observations with miR-184 reduction of Nrx-IV and Mcr in the wing disc [26]. However, the impact of miR-184 expression on the pSJ structure integrity differed between the two tissues. In the wing disc, overexpressing miR-184 did not affect the pSJ domain structure or polarity or epithelial integrity [26]. However, in glia, miR-184 overexpression resulted in disrupted pSJ phenotypes, ranging from intermittent breaks with some pSJ remnants to a complete loss of the pSJ structure. This tissue-specific differences could be explained by the variations in the 3' UTR regions of these pSJ proteins, which can affect their sensitivity to miR-184. Such differences could also account for the varying impact of miR-184 on TCJ structure in glia compared to wing imaginal discs. For instance, the *Gli* gene has two distinct 3' UTR isoforms—one containing the miR-184 target site and one without [26]. While miR-184 expression caused only a modest reduction in *Gli* levels in the SPG, it led to a complete loss of *Gli* in the wing disc [26]. This suggests that the shorter *Gli* mRNA isoform, which lacks the miR-184 target sequence, may be the predominant form expressed in the SPG.

Of the pSJ proteins targeted by miR-184, the defects we observed could be due to the loss of one of the proteins as pSJ proteins are interdependent for localization [9,11,13,15,54]. This interdependency of pSJ proteins makes it hard to interpret which of these proteins is directly regulated by miR-184 or if a combination of loss of Nrx-IV, Kune, Sinu and/or Mcr is needed for the pSJ disruption. The low levels of Sinu in the brain lobe pSJ was surprising in that loss of Sinu leads to blood-brain barrier defects in embryos [3]. The much stronger expression of Sinu in the peripheral pSJ compared to Kune suggests that there might be differential expression and functions of the claudin-like proteins in the CNS versus PNS pSJ. Sinu in the peripheral glia was by far the most sensitive to miR-184 overexpression, showing an almost complete loss of its expression. Sinu, along with other core pSJ proteins, is essential for proper organization of mature, immobile septate junctions, where loss of any one of the core proteins leads to increased turnover of pSJ proteins [54]. Thus, Sinu may be the key regulatory point for miR-184, given that this was the only protein we found strongly downregulated. Notably, in all the cases, the remnants of Kune and Mcr consistently colocalized with those of Nrx-IV, suggesting that Kune, Mcr, and Nrx-IV proteins are more stable (i.e., have a low turnover rate) or that these mRNAs are not targeted by miR-184 in the glia, and have a change in expression pattern due to loss of Sinu or another SJ protein. However, given the strong interdependence of the pSJ protein complex it is likely that there is not one key target but rather the combined downregulation of multiple proteins that leads to pSJ disruption.

Since SPG are few in number and the morphology is too thin to isolate, we could not quantify the protein expression levels of pSJs in the SPG specifically to determine which proteins are most strongly affected. Surprisingly, while protein expression levels were affected for the predicted miR-184 pSJ proteins, their mRNA levels were not. Our qRT-PCR results showed that miR-184 overexpression did not affect the mRNA levels of any of the predicted target mRNAs. The lack of reduction in pSJ mRNA levels is in contradiction to *in vitro* studies that identified miR-184 as a potential regulator of pSJ mRNAs [25]. These observations can be attributed to several underlying reasons. It is possible that miR-184 does not target any of the tested pSJ mRNAs but targets other proteins crucial for the SJ protein and junctional stability. Other potential predicted pSJ targets that were not tested in this study include Transferin2, Undicht, Pasiflora, and Crooked (targetscanfly). In future, the expression pattern and mRNA levels of these other miR-184 targets could be studied. Another explanation to these results might be owed to the mRNA isolation method. We isolated mRNA for the qRT-PCR analysis from brain lobes and the ventral nerve cord, assuming that SPG are the only cells that express pSJ mRNAs/proteins.

 

However, since pSJs proteins are known for their non-junctional roles in different tissues and at different developmental stages [8,12,13,15,55–58], it is possible that other glial cells or neurons might express these pSJ proteins. Thus, our qRT-PCR analysis might not accurately reflect the SJ mRNA levels in SPG. However, our observations that both the Nrv2 mRNA and protein levels increased does suggest that our approach was able to detect changes in the SPG expression levels. Finally, it is possible that miR-184 binds to target mRNAs and affects their protein level by translational repression rather than through mRNA degradation. Prior research has demonstrated that miRNAs can repress translation by sequestering target mRNAs and blocking the initiation of translation [59]. For instance, the posterior determinant protein, Oskar is regulated by miRNA-312 by translational repression [60–62], and miR-2 targets its mRNAs through inhibition of translation initiation and potential recruitment to P-bodies [63,64]. While a similar mechanism has not been reported in glial cells, miR-184 might be translationally repressing the pSJ mRNAs in subperineurial glia.

Overall, our studies showed that a specific miRNA, miR-184, is differentially expressed in tissues forming smooth SJs versus pleated SJs, with a lack of miR-184 in tissues that express pleated SJs. This variation could indicate different levels of regulation to ensure the expression of the distinct protein components that form the different types of septate junctions. The lack of miR-184 in these tissues is critical as we found that expression in subperineurial glia leads to loss of pSJ integrity and disruption of permeability barriers but without affecting the mRNA levels of targeted pSJ components. Thus, mRNA-184 can regulate multiple pleated septate junction proteins either directly through loss of translation or indirectly by disruption of the septate junction domain.

## Supporting information

**S1 Fig. Predicted miRNA target sites in pSJ and TCJ mRNAs.** Conserved miRNA sites in the 3' UTR of Nrx-IV, sinu, kune, Mcr, Gli and M6 predicted by target scan (www.targetscan.org/fly_72/) are shown. miR-184 target sequences are indicated in red.
(TIF)

**S1 File. Values for all quantifications.**
(XLSX)

## Acknowledgments

*Drosophila melanogaster* stocks were obtained from the Bloomington Drosophila Stock Center and the Vienna Drosophila Resource Center and monoclonal antibodies were obtained from the Developmental Studies Hybridoma Bank. We extend our sincere thanks to Dr. Mary Gilbert for her assistance and Drs. Greg Beitel, Robert Ward, and Stefan Luschnig for generously sharing of antibodies. We also wish to thank Dr. Douglas Allan for providing the NLS-YFP plasmid and Dr. Elizabeth Rideout for access to the qPCR facility. Optical imaging was made possible through the Life Sciences Institute Imaging (LSI Imaging) facility at the University of British Columbia, Vancouver, Canada.

## Author contributions

**Conceptualization:** Vanessa J. Auld, Sravya Paluri.

**Formal analysis:** Sravya Paluri.

**Funding acquisition:** Vanessa J. Auld.

**Investigation:** Sravya Paluri.

**Supervision:** Vanessa J. Auld.

**Visualization:** Sravya Paluri.

**Writing – original draft:** Sravya Paluri.

**Writing – review & editing:** Vanessa J. Auld, Sravya Paluri.

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
