## [Decision Letter · Decision Letter 0]

14 Aug 2025

Dear Dr. Auld,

Thank you for submitting your manuscript to PLOS ONE. After careful consideration, we feel that it has merit but does not fully meet PLOS ONE’s publication criteria as it currently stands. Therefore, we invite you to submit a revised version of the manuscript that addresses the points raised during the review process.

Both reviewers recommend minor changes in your article before is ready for publication. Please, address all the comments in your rebuttal letter.

We look forward to receiving your revised manuscript.

Kind regards,

Carlos Oliva, PhD

Academic Editor

PLOS ONE

Journal Requirements:

2. Thank you for stating the following financial disclosure: [V.A AWD-010749 Natural Sciences and Engineering Research Council]. 

3. We note that your Data Availability Statement is currently as follows: [All relevant data are within the manuscript and its Supporting Information files]

Reviewers' comments:

Reviewer's Responses to Questions

**Comments to the Author**

1. Is the manuscript technically sound, and do the data support the conclusions?

Reviewer #1: Yes

Reviewer #2: Yes

2. Has the statistical analysis been performed appropriately and rigorously?

Reviewer #1: Yes

Reviewer #2: Yes

3. Have the authors made all data underlying the findings in their manuscript fully available?

Reviewer #1: Yes

Reviewer #2: Yes

4. Is the manuscript presented in an intelligible fashion and written in standard English?

Reviewer #1: Yes

Reviewer #2: Yes

Reviewer #1: The manuscript by Paluri and Auld investigates the role of miR-184 as a regulator of pleated septate junctions (pSJs) during the development of the Drosophila blood-brain barrier (BBB). The authors characterize miR-184 expression using a transgenic fluorescent reporter and show that it is restricted to tissues that do not form pSJs. In contrast, ectopic expression of miR-184 in subperineurial glia disrupts the localization of pSJ proteins in both the central nervous system and peripheral nerves. Interestingly, this mislocalisation occurs without a detectable decrease in the mRNA levels of core pSJ components. Instead, they observe an increase in nrv2 transcripts, a phenotype that can be phenocopied by RNAi-mediated downregulation of neurexin-IV.

Overall, this is a well-executed and thoughtfully written study. The manuscript is clear and the discussion addresses most of the relevant limitations of the experimental approaches. While the conclusions may be of limited impact—given that miR-184 is not endogenously expressed in the Drosophila BBB—this work provides a detailed description of miR-184 expression and its potential role in preventing ectopic expression of pSJ components. I recommend acceptance of the manuscript with minor revisions.

Minor comments.

1. I suggest including a figure that shows the miR-184 binding sites in Nrx-IV used for generating the NLS-YFP sensor construct. It would also be of interest to indicate predicted or potential miR-184 binding sites in other core pSJ components.

2. While the manuscript focuses on larval tissues, it would strengthen the study to analyze miR-184 expression in the adult BBB to rule out expression later in development.

3. The authors report that misexpression of miR-184 using SPG-GAL4 (moody-GAL4) is lethal. However, this driver is also known to be active in other tissues such as the midgut, salivary glands, and fat body. To ensure that the observed lethality results specifically from BBB defects, the experiment should be repeated using mdr65-GAL4 (R54C07), which has more restricted expression in subperineurial glia.

4. It would be interesting to assess whether co-expression of UAS-Nrx-IV lacking the 3′UTR (thus evading miR-184 regulation) together with miR-184 can rescue the loss of other pSJ components such as Kune and Mcr. This would strengthen the conclusion that Nrx-IV is a primary functional target of miR-184 in this context.

Reviewer #2: 

The paper by Sravya Paluri and Vanessa J. Aull is an impeccable descriptive study of the expression of microRNA miR-184. In it, the authors employ appropriate tools to investigate its expression in larval tissues, identifying the specific tissues and cells where the microRNA is expressed. They then focus on the brain, particularly on the glia forming the blood-brain barrier (BBB), the cells of the SPG. In these cells, they describe the effects of downregulating the microRNA on the morphology and permeability of the barrier, as well as on the abundance of various proteins associated with pleatet septate junctions (pSJs), which are key cellular components in forming the barrier in SPG cells. They found that miRNA downregulation disrupts barrier formation, increasing its permeability, alters the expression of several pSJ proteins, and affects larval locomotion. Notably, it does not alter the mRNA levels of the proteins that constitute the SJs.

However, they observed that the NRV2 protein, a component of the pSJ but not a direct target of the microRNA, had increased levels, suggesting a regulatory relationship among pSJ proteins.

Although mainly descriptive, the results are of interest to the community working on microRNAs and septate junctions. While it is not surprising that proteins forming part of a larger complex such as the SJ might increase their expression as a compensatory mechanism, this phenomenon has been previously described in synapses, particularly in the postsynaptic density, among other contexts.

Minor comments:

- It could be interpreted that the only target of miR-148 is septate junction proteins. Whether this is the case or not, it would be important to clarify this point in the introduction.

- Why is the Sinu label in the brain's SPG so faint, while it is clear in the SPG of peripheral nerves? Does this suggest that the SPG of peripheral nerves has a different type of SJ?

**Do you want your identity to be public for this peer review?** For information about this choice, including consent withdrawal, please see our Privacy Policy

Reviewer #1: No

Reviewer #2: No

---

## [Author Response · Author response to Decision Letter 1]

22 Oct 2025

Please find below our response to the reviewers.

Thanks to the reviewers for their supportive comments and supportive suggestions. We have outlined the revisions and responses to both reviewers below.

Reviewer #1:

Minor comments.

1. I suggest including a figure that shows the miR-184 binding sites in Nrx-IV used for generating the NLS-YFP sensor construct. It would also be of interest to indicate predicted or potential miR-184 binding sites in other core pSJ components.

We have included a new figure as supplemental (S1 Fig) showing the predicted miRNA target sequence in Nrx-IV and other core pSJ mRNAs. The actual nucleotide sequence used is included in the materials and methods.

2. While the manuscript focuses on larval tissues, it would strengthen the study to analyze miR-184 expression in the adult BBB to rule out expression later in development.

This is a very good idea however the focus of this manuscript was on the larval expression and levels. This was mostly because our expertise (using the Gal4 drivers and pSJ markers) and the majority of literature is focused on the 3rd instar stages thus allowing us to better interpret the effects miRNA-184 in this context.

3. The authors report that misexpression of miR-184 using SPG-GAL4 (moody-GAL4) is lethal. However, this driver is also known to be active in other tissues such as the midgut, salivary glands, and fat body. To ensure that the observed lethality results specifically from BBB defects, the experiment should be repeated using mdr65-GAL4 (R54C07), which has more restricted expression in subperineurial glia.

This is an excellent idea and we obtained mdr65-Gal4 line and carried out the experiment. Using this driver we observed the same degree of larval lethality observed with the SPG-Gal4 driver and have now included this data in the manuscript.

4. It would be interesting to assess whether co-expression of UAS-Nrx-IV lacking the 3′UTR (thus evading miR-184 regulation) together with miR-184 can rescue the loss of other pSJ components such as Kune and Mcr. This would strengthen the conclusion that Nrx-IV is a primary functional target of miR-184 in this context.

We do not think that Nrx-IV is the primary functional target of miR-184 and have revised the manuscript to make this clearer (lines 826-828). That being said this is an excellent suggestion and we did try in the process of this study to do exactly this experiment. But for technical reasons we were unable to obtain a clean copy of the UAS-Nrx-IV transgene including an epitope tag.

Reviewer #2:

Minor comments:

- It could be interpreted that the only target of miR-148 is septate junction proteins. Whether this is the case or not, it would be important to clarify this point in the introduction.

We have clarified in the introduction (lines 73-74) that there are many other miR-184 targets and specifically noted that only 10 are predicted pSJ mRNAs.

- Why is the Sinu label in the brain's SPG so faint, while it is clear in the SPG of peripheral nerves? Does this suggest that the SPG of peripheral nerves has a different type of SJ?

Yes we feel that the claudin-like proteins are likely differential expressed in the CNS versus the PNS. We have emphasized this point the discussion (lines 802-804)

---

## [Decision Letter · Decision Letter 1]

17 Nov 2025

microRNA-184 distribution and consequences on glial septate junctions and the blood-brain barrier

PONE-D-25-37097R1

Dear Dr. Auld,

We’re pleased to inform you that your manuscript has been judged scientifically suitable for publication and will be formally accepted for publication once it meets all outstanding technical requirements.

Kind regards,

Carlos Oliva, PhD

Academic Editor

PLOS ONE

Additional Editor Comments (optional):

Reviewers' comments:

Reviewer's Responses to Questions

**Comments to the Author**

Reviewer #1: All comments have been addressed

Reviewer #2: All comments have been addressed

2. Is the manuscript technically sound, and do the data support the conclusions?

Reviewer #1: Yes

Reviewer #2: Yes

3. Has the statistical analysis been performed appropriately and rigorously?

Reviewer #1: Yes

Reviewer #2: Yes

4. Have the authors made all data underlying the findings in their manuscript fully available?

Reviewer #1: Yes

Reviewer #2: Yes

5. Is the manuscript presented in an intelligible fashion and written in standard English?

Reviewer #1: Yes

Reviewer #2: Yes

Reviewer #1: (No Response)

Reviewer #2: The revised manuscript have addressed all the comments and I do not have further questions. THe work is sound and well performed.

**Do you want your identity to be public for this peer review?** For information about this choice, including consent withdrawal, please see our Privacy Policy

Reviewer #1: No

Reviewer #2: No

---

## [Editor Report · Acceptance letter]

PONE-D-25-37097R1

PLOS ONE

Dear Dr. Auld,

I'm pleased to inform you that your manuscript has been deemed suitable for publication in PLOS ONE. Congratulations! Your manuscript is now being handed over to our production team.

Kind regards,

on behalf of

Dr. Carlos Oliva

Academic Editor

PLOS ONE